# Triple-wavelength depolarization-ratio profiling of Saharan dust over Barbados during SALTRACE in 2013 and 2014

Moritz Haarig[1], Albert Ansmann[1], Dietrich Althausen[1], André Klepel[1,2], Silke Groß[3], Volker Freudenthaler[4], Carlos Toledano[5], Rodanthi-Elisavet Mamouri[6], David A. Farrell[7], Damien A. Prescod[7], Eleni Marinou[8], Sharon P. Burton[9], Josef Gasteiger[10], Ronny Engelmann[1], and Holger Baars[1]

[1]Leibniz Institute for Tropospheric Research, Leipzig, Germany
[2]Goldschmidt Thermit GmbH, Technology Innovation Center, Leipzig, Germany
[3]German Aerospace Center, Institute of Atmospheric Physics, Oberpfaffenhofen, Germany
[4]Ludwig Maximilians University, Meteorological Institute, Munich, Germany
[5]University of Valladolid, Group of Atmospheric Optics, Valladolid, Spain
[6]Cyprus University of Technology, Department of Civil Engineering and Geomatics, Limassol, Cyprus
[7]Caribbean Institute for Meteorology and Hydrology, Bridgetown, Barbados
[8]Inst. for Astronomy, Astrophysics, Space Appl. and Remote Sensing, National Observatory Athens, Athens, Greece
[9]NASA Langley Research Center, MS 475, Hampton, VA 23681, USA
[10]University of Vienna, Aerosol Physics and Environmental Physics, Vienna, Austria

*Correspondence to:* Moritz Haarig (haarig@tropos.de)

**Abstract.** Triple-wavelength polarization lidar measurements in Saharan dust layers were performed at Barbados (13.1°N, 59.6°W), 5000–8000 km west of the Saharan dust sources, in the framework of the Saharan Aerosol Long-range Transport and Aerosol-Cloud-Interaction Experiment (SALTRACE-1, June-July 2013, SALTRACE-3, June-July 2014). Three case studies are discussed. High quality was achieved by comparing the dust linear depolarization ratio profiles measured at 355, 532, and 1064 nm with respective dual-wavelength (355, 532 nm) depolarization ratio profiles measured with a reference lidar. A unique case of long-range transport dust over more than 12000 km is presented. Saharan dust plumes crossing Barbados were measured with an airborne triple-wavelength polarization lidar over Missouri in the Midwestern United States 7 days later. Similar dust optical properties and depolarization features were observed over both sites indicating almost unchanged dust properties within this one week of travel from the Caribbean to the United States. The main results of the triple-wavelength polarization lidar observations in the Caribbean in the summer seasons of 2013 and 2014 are summarized. On averaged, the particle linear depolarization ratios for aged Saharan dust were found to be 0.252±0.030 at 355 nm, 0.280±0.020 at 532 nm, and 0.225±0.022 at 1064 nm after approximately one week of transport over the tropical Atlantic. Based on published simulation studies we present an attempt to explain the spectral features of the depolarization ratio of irregularly shaped mineral dust particles, and conclude that most of the irregularly shaped coarse-mode dust particles (particles with diameters $>1$ $\mu$m) have sizes around 1.5–2 $\mu$m. The SALTRACE results are also set into the context to the SAMUM-1 (Morocco, 2006) and SAMUM-2 (Cabo Verde, 2008) depolarization ratio studies. Again, only minor changes in the dust depolarization characteristics were observed on the way from the Saharan dust sources towards the Caribbean.

# 1 Introduction

Mineral dust belongs to the major natural atmospheric aerosol components and influences weather and climate, visibility, air quality, and human health. Large efforts are undertaken to develop sophisticated dust transport models to provide predictions of dust occurrence, vertical distribution of dust particles and their impact on the Earth's radiation field, cloud formation, and environmental/air-quality conditions (Huneeus et al., 2011; Heinold et al., 2011; Garimella et al., 2014; Solomos et al., 2017). A variety of features of the impact of dust on (climate-relevant) atmospheric processes are not well understood and represented in atmospheric models, and thus need to be explored, preferably in comprehensive field campaigns such as Aerosol Characterization Experiment ACE-Asia (Huebert et al., 2003; Shimizu et al., 2004), the Puerto Rico Dust Experiment PRIDE (Colarco et al., 2003; Reid et al., 2003), the Saharan Dust Experiment SHADE (Tanré et al., 2003), the Saharan Mineral Dust Experiments SAMUM-1 (Heintzenberg, 2009) and SAMUM-2 (Ansmann et al., 2011), the Dust and Biomass-burning Experiment DABEX (Haywood et al., 2008), the Dust Outflow and Deposition to the Ocean project DODO (McConnell et al., 2008), the Geostationary Earth Radiation Budget Intercomparisons of Long-wave and Short-wave radiation GERBILS (Johnson and Osborne, 2011), Fennec (Ryder et al., 2013), the Saharan Aerosol Long-Range Transport and Aerosol-Cloud-Interaction Experiment SALTRACE (Weinzierl et al., 2017) and the Study of Saharan Dust Over West Africa SHADOW (Veselovskii et al., 2016).

Besides a precise description of dust in atmospheric models, there is also a strong need for a better knowledge of the link between the microphysical and optical dust properties to improve dust profiling and retrieval techniques (Veselovskii et al., 2010, 2016; Müller et al., 2013; Lopatin et al., 2013; Chaikovsky et al., 2016; Bovchaliuk et al., 2016). Recently, a new aerosol retrieval technique was proposed, which makes use of depolarization-ratio profiling and permits the separation of fine dust, coarse dust, and residual (marine or anthropogenic) aerosol profiles in terms of light backscatter, extinction, and mass concentration (Mamouri and Ansmann, 2014). Based on SALTRACE triple-wavelength polarization measurements, it is investigated at which of the three wavelengths this new method works best and the uncertainties are lowest Mamouri and Ansmann (2017).

One of the fundamental open questions regarding the influence of mineral dust on climate is related to the specific impact of the size, shape, and chemical composition characteristics of desert dust particles on light scattering and depolarization (Dubovik et al., 2006; Wiegner et al., 2009; Müller et al., 2010b, a, 2012; Gasteiger et al., 2011; Kemppinen et al., 2015a, b). Besides modeling studies and field observations, laboratory experiments contribute to this field of research (West et al., 1997; Volten et al., 2001; Liu et al., 2003; Curtis et al., 2008), recently also with focus on lidar applications (Sakai et al., 2010; David et al., 2013; Miffre et al., 2016; Järvinen et al., 2016). Although significant progress is made during the last decade, models describing the scattering properties of desert dust particles from forward scattering to backward scattering (up to angles of exactly 180°) need a number of assumptions especially about the particle morphology and composition when explaining the optical effects gained from active and passive remote sensing at different wavelength from the UV to IR (Gasteiger et al., 2011; Kemppinen et al., 2015a, b). Observations of dust optical properties are in strong contradiction with model simulations when a spherical dust particle shape model is applied. The widely used and accepted approach to describe dust particles as spheroids works well in the case of sun photometer retrievals (Dubovik et al., 2006). However, significant uncertainties in the observa-

tional products arise whenever lidar measurements (and thus 180° scattering processes) come into play and are included in the data analysis in which a spheroidal shape model is used (Wagner et al., 2013; Veselovskii et al., 2010, 2016; Müller et al., 2013). There is a clear need for complex efforts of simulation studies, laboratory investigations, and field observations in order to better parametrize the relationship between the dust particle shape and size distribution characteristics and the scattering phase function with emphasis on depolarization and scattering at high scattering angles. The depolarization ratio observed with lidar is rather sensitive to the dust particle shape (Gasteiger et al., 2011; Kemppinen et al., 2015a). The spectral dependence of depolarization, as presented in this article, contains in addition information on the dust size spectrum. Thus triple-wavelength depolarization ratio observations are of great value for the optical modeling community (Gasteiger and Freudenthaler, 2014) and support efforts to develop realistic dust shape models, which are not available yet. In the first stage, however we have to demonstrate that triple-wavelength depolarization observations can be successfully performed.

To contribute to this field of dust research, we re-designed and upgraded our multiwavelength polarization/Raman lidar BERTHA (Backscatter Extinction lidar-Ratio Temperature Humidity profiling Apparatus) (Althausen et al., 2000; Tesche et al., 2009; Haarig et al., 2016b). BERTHA has been used in nine field campaigns in Europe, Asia, and Africa from 1997–2008 (Wandinger et al., 2002; Ansmann et al., 2002; Franke et al., 2003; Tesche et al., 2009, 2011a). We implemented new channels to permit simultaneous observations of dust linear depolarization ratios at 355, 532, and 1064 nm. Freudenthaler et al. (2009) and Burton et al. (2015) showed already that the dust linear depolarization ratio significantly changes with the transmitted laser wavelength, obviously as a result of changing contributions of fine-mode dust particle (particles with diameters $<1$ $\mu$m) and coarse-mode (super-micrometer) particles to the light depolarization. Burton et al. (2015) recently presented triple-wavelength polarization lidar observations in an aged and fresh dust layer performed with an airborne high spectral resolution lidar (HSRL-2). The main goal of the paper is to present for the first time ground-based triple-wavelength polarization/Raman lidar observations (case studies) of the depolarization ratio of Saharan dust after long range transport and to provide a high-quality statistical data set of dust depolarization ratios at 355, 532, and 1064 nm. We conducted three campaigns on the Caribbean island of Barbados in the framework of SALTRACE. Two of them took place in the summer seasons of 2013 (SALTRACE-1) and 2014 (SALTRACE-3) to study Saharan dust layers advected from Africa towards North America. The third field campaign (SALTRACE-2) was performed in February-March 2014 to continue our research on mixed dust and smoke transport towards America during the winter half year (Ansmann et al., 2009; Baars et al., 2011; Tesche et al., 2011b; Rittmeister et al., 2017).

We begin our SALTRACE report with a short description of the SALTRACE experiment and instrumentation. The main part of Sect. 2 (and the Appendix) deals with the explanations of the triple-wavelength polarization lidar BERTHA. The results are presented in Sects. 3 and 4. In Sect. 3, three case studies are discussed. A statistical overview of the SALTRACE depolarization measurements is given in Sect 4. The findings are compared with respective results from the foregoing SAMUM-1 and SAMUM-2 field campaigns and the Cloud-Aerosol Lidar with Orthogonal Polarization (CALIOP) satellite observations in the discussion Section 5. Concluding remarks are given in Sect. 6.

## 2 SALTRACE campaigns and instrumentation

### 2.1 The SALTRACE project

The SALTRACE field campaigns performed in the summer of 2013 and in the winter and summer of 2014 belong to the SAMUM-SALTRACE field campaign series. As shown and illustrated in Fig. 7 of Weinzierl et al. (2017), six comprehensive dust field campaigns have been conducted since 2006: SAMUM-1, SAMUM-2a and 2b, and SALTRACE-1, 2, and 3. The Saharan Mineral Dust Experiment (SAMUM) project SAMUM-1 (Heintzenberg, 2009) took place in southern Morocco (May-June 2006) to investigate the role of freshly emitted dust in the climate system. SAMUM-2 (Cabo Verde, SAMUM-2a, January-February 2008, SAMUM-2b, May-June 2008) (Ansmann et al., 2011) investigated the dust physico-chemical, optical, and radiative properties of mixtures of biomass burning smoke and mineral dust (SAMUM-2a, winter transport regime) and of pure dust (SAMUM-2b, summer transport mode) after an atmospheric transport over 1000-3000 km (1-3 days after emission). During SALTRACE, we investigated the dust properties after an atmospheric travel over 5–12 days and 5000–8000 km (Weinzierl et al., 2017).

As the logistically favorable field site for lidar observations we selected the Caribbean Institute of Meteorology and Hydrology (CIMH, 13.1°N, 59.6°W, 90 m above sea level) in Husbands, in the northern parts of the capital Bridgetown at the west coast of Barbados. The station is not influenced by any local (island) anthropogenic pollution because of the steady northeasterly airflow and the absence of pollution sources in the northern part of Barbados upwind the lidar station. In the summer months of June and July (SALTRACE-1 and 3), transported dust layers were observed. The SALTRACE lidar activities were complemented by shipborne observations along the main Saharan dust transport route over the tropical North Atlantic in April-May 2013 (Kanitz et al., 2014; Rittmeister et al., 2017; Ansmann et al., 2017), in situ observations of microphysical (size distribution, mass concentration, particle shape), chemical, and optical aerosol properties at Ragged Point (20 km east of CIMH) at the east coast of Barbados (Kristensen et al., 2016; Wex et al., 2016) and airborne in situ aerosol observations and Doppler lidar measurements of aerosol layering and atmospheric wind fields (Chouza et al., 2015, 2016a, b; Weinzierl et al., 2017). The SALTRACE in situ observations include studies of the efficiacy of aged desert dust to serve a cloud condensation nuclei (Kristensen et al., 2016; Wex et al., 2016; Weinzierl et al., 2017), and modeling studies of dust transport across the Atlantic (Chouza et al., 2016a; Ansmann et al., 2017) and the impact of the Caribbean island on the airflow and downward mixing of dust (Jähn et al., 2016; Chouza et al., 2016b).

### 2.2 Meteorological Conditions

In Rittmeister et al. (2017), the dust transport from Africa towards the Caribbean is discussed. The main features of dust layering across Atlantic described by the conceptual model (Karyampudi et al., 1999) are illuminated and compared with the shipborne SALTRACE lidar observations. According to the conceptual model (Karyampudi et al., 1999) hot, dry, dust-laden air masses emerge from the western coast of Africa as a series of large-scale pulses in the summer months. Associated with easterly wave activity, Saharan dust outbreaks occur as discrete episodic pulses, which generally last 3–5 days. These dust outbreaks are mostly confined to a well-mixed layer, the Saharan air layer (SAL), that often extends to 5–6 km in height over

West Africa due to intense solar heating in summer months. The airborne dust is carried westward by the prevailing easterly flow in the latitude belt of 10°–25°N. As the dust plumes are advected further west in the predominantly easterly flow, the base of the SAL rises rapidly as it is undercut by the relatively clean northeasterly trade winds. The well-mixed SAL resides above the trade wind inversion layer which is on top of the marine aerosol layer (MAL). The dust transport takes usually 5–7 days across the Atlantic. The strong temperature inversion at the base of the SAL limits convective activity and consequently precludes the possibility of strong wet deposition, except during periods with deep convection and precipitation.

## 2.3 Triple-wavelength polarization lidar BERTHA

We begin with a short historical overview of polarization lidar observations of tropospheric dust. The polarization lidar technique was applied to tropospheric aerosols for the first time in the early 1970s (McNeil and Carsweil, 1975). Systematic measurements of the depolarization ratio in desert dust layers started in the 1980s in eastern Asia (Kobayashi et al., 1985; Iwasaka et al., 1988) and demonstrated the importance of polarization lidar for dust monitoring. Consequently, the Asian Dust Lidar Network was established in the 1990s (Murayama et al., 2001; Shimizu et al., 2004). In Europe, systematic Saharan dust studies with polarization lidars began in the 1990s (Gobbi, 1998; Gobbi et al., 2000; di Sarra et al., 2001). Dust was investigated later on by means of polarization lidar in the framework of the European Aerosol Research Lidar Network (e.g., Ansmann et al., 2003)). All of the ground-based observations, conducted before SALTRACE, were based on single-wavelength lidar observations, although several single-wavelength-lidar systems were combined during dedicated field campaigns such as SAMUM-1 and 2 (Freudenthaler et al., 2009; Groß et al., 2011b; Tesche et al., 2011a). In the majority of applications the laser wavelength was 532 nm. Sugimoto and Lee (2006) presented the first ground-based dual-wavelength polarization lidar observations of dust performed at 532 and 1064 nm. Airborne dual-wavelength polarization lidar observations (at 532 and 1064 nm) were realized in Saharan dust aboard the Falcon of the Germany Aerospace Center during SAMUM-1 in 2006 (Freudenthaler et al., 2009). During SALTRACE, dual-wavelength polarization lidars were operated simultaneously at 355 and 532 nm (Kanitz et al., 2014; Groß et al., 2015; Rittmeister et al., 2017). And recently the first triple-wavelength polarization lidars were developed and performed dust observations at 355, 532, and 1064 nm aboard an aircraft (Burton et al., 2015) and at ground (Haarig et al., 2016a).

The schematic structure of the lidar system BERTHA is shown in Fig. 1. Two Nd:YAG lasers transmit linearly polarized laser pulses at 355 and 1064 nm (first laser) and at 532 nm (second laser) with a repetition rate of 30 Hz. Two linear polarizers are installed behind the laser and before the beam expander to further clean the polarization of the outgoing light. The pulse energies can be as high as 1000 mJ (1064 nm), 800 mJ (532 nm) and 120 mJ (355 nm) in the ideal case of well-working optical elements in the transmission unit of the lidar. However, the pulse energies were only about 50% of these maximum values during the SALTRACE campaigns. Two lasers are used for two reasons, firstly to have a frequency stabilized 532 nm laser for the implementation of the HSRL-channel and secondly to have a backup laser in the field campaign. The laser beams are aligned on an optical axis and directed through a beam expander. The beams are expanded tenfold and afterwards pointed into the atmosphere at an off-zenith angle of 5°. By using a tilt angle of $> 3°$ specular reflection by falling and horizontally aligned ice crystals does no longer influence the measurements as our experience shows.

A Cassegrain telescope with 53 cm diameter collects the backscattered light. The receiver field of view is 0.8 mrad. To avoid overloading of the photomultipliers (PMTs, photon counting mode) in the near range we restricted the maximum count rate to 20 MHz for the signal maximum in about 500 m height by means of neutral density filters.

The receiver unit was completely re-designed to measure the linear depolarization ratio at all three wavelengths. In addition, a high spectral resolution (HSR) channel was added (see Fig. 1). All detectors are photomultiplier tubes (PMT) working in the photon counting mode (H10721P-110 from Hamamatsu). But for the 1064 nm channels the PMT R3236 from Hamamatsu are used at a temperature below $-30°C$ to reduce signal noise in the near infrared. The elastic backscatter signals (total signals) as well as the so-called cross-polarized signal components are detected at the three emitted wavelengths (355, 532, 1064 nm). Polarization filters, each adjusted orthogonal to the plane of linear polarization of the outgoing laser pulses, are placed in front of the detectors which enable the detection of the cross-polarized laser radiation at the three wavelengths. For these six elastic channels, interference filters with 1 nm FWHM (full width at half maximum) are placed in front of each PMT. The vibrational-rotational Raman signals at 387 and 607 nm (nitrogen) and 407 nm (water vapor, night time only) are detected as well. Interference filters with 3 nm FWHM are used at the Raman wavelengths. A double-grating monochromator enables the detection of pure rotational Raman signals from nitrogen and oxygen (J0, J6 and J12) from the 532 nm emission wavelength. Neutral density filters are placed in front of each detector to adjust the signal to the linear range of the detector to avoid dead time effects of the photo multipliers. Nevertheless the PMTs are tested in the lab to correct high counting rates for dead time effects if necessary (Engelmann et al., 2016).

The signals are detected with a range resolution of 7.5 m and a time resolution of 5 to 30 s. A camera is used to visualize the overlap between the 532 nm laser beam and the receiver field of view. The camera is permanently placed in the position of a receiving channel (see Engelmann et al., 2016). Complete overlap is reached at 800–1000 m for the 532 nm related channels and approximately at 1500 m for the 355 nm-related channels. The 355 and 532 nm backscatter coefficient is derived from the ratio of the elastic backscatter signal to the respective nitrogen Raman signal (387 or 607 nm) and therefore not much affected by the overlap profile. Even at 1064 nm we used the nitrogen Raman signal at 387 nm as the reference signal in the 1064 nm backscatter retrieval. Both signals are caused by backscattered laser-1 photons (see Fig. 1) so that almost the same overlap characteristics holds for both signals and the overlap effects widely cancels out when forming the signal ratio. We used the sun photometer observations (Sect. 2.6) of the spectral slope of aerosol extinction for the correction of minor differential (387 nm vs 1064 nm) particle extinction effects in the backscatter retrieval. On clear (dust-free) days we checked the overlap function according to Wandinger and Ansmann (2002). Trustworthy results can be obtained at altitudes above about 400 m for the backscatter coefficient and above about 1000 m for the extinction coefficient, derived from the 387 and 607 nm nitrogen Raman signal profiles.To control the correct alignment, a telecover test (Freudenthaler, 2008) with 8 segments (4 inner and 4 outer) has been performed during SALTRACE-2 and at Leipzig in the autumn of 2014.

The focus of this article is on depolarization-ratio observations. In order to ensure a high quality of the data, the polarization sensitivity of the lidar system was characterized carefully. The polarization sensitive transmission of the optical elements in the emitter and the receiver has been characterized. A detailed description of the polarization characterization can be found in Appendix A.

The basic lidar-derived quantity is the volume linear depolarization ratio defined as the ratio of cross- to co-polarized signal components (Gimmestad, 2008). Co and cross denote the planes of polarization (for which the receiver channels are sensitive) parallel and orthogonal to the plane of linear polarization of the transmitted laser pulses, respectively. In the case of BERTHA we measure the cross-polarized and total (cross + co-polarized) signal components and thus determined the co-polarized signal component from the cross-polarized and total signal components (more details are given in the Appendix).

The volume depolarization ratio at 355 and 532 nm is influenced by light depolarization by air molecules, aerosol, and cloud particles. To obtain the particle depolarization ratio a correction for molecular depolarization effects have to be applied (Biele et al., 2000). To account for molecular backscatter, extinction and depolarization contributions to the measured lidar signals we used the SALTRACE radiosonde observations at CIMH, which were performed twice a day (Sect. 2.7). The radiosonde height profiles of air temperature and pressure profiles permit the computation of the actual height profile of air molecule number concentration over Barbados.

The presence of cirrus is routinely used to check the consistence of the depolarization ratios at different wavelengths among each other. Because the size of the ice crystals is usually much larger than the laser wavelength, the measured optical properties are close together for the 355 to 1064 nm wavelength range. This can be used to evaluate the quality of the depolarization ratio observations for each of the three wavelengths. Case study II in the next section will be an example for these routine checks.

An extended analysis of systematic uncertainties in the retrieved optical properties can be found in Freudenthaler et al. (2009); Freudenthaler (2016); Tesche et al. (2009, 2011a, b); Bravo-Aranda et al. (2016). The error bars of the retrieval products given in the next sections show standard deviations considering the overall uncertainty.

## 2.4   POLIS

During the SALTRACE-1 campaign in the summer of 2013, the dual-wavelength polarization lidar POLIS (Portable Lidar System) of the Munich University (Groß et al., 2015) was operated at CIMH, about 50 m north of the BERTHA lidar. POLIS is a well-designed and characterized six-channel polarization/Raman lidar and provides volume linear depolarization-ratio profiles at 355 and 532 nm with high accuracy (Freudenthaler et al., 2016; Bravo-Aranda et al., 2016). POLIS is used as the reference polarization lidar system in EARLINET calibration and quality-assurance activities. The deployment of both lidars at the same site was motivated by the fact that the 13-channel BERTHA lidar, which integrates the HSR lidar technique, Raman aerosol, water vapor, and temperature profiling methods, and now in addition the multiwavelength depolarization-ratio profiling option in one system, is a complex lidar system with a large number of potential sources for uncertainties (see Appendix A). Therefore to avoid any risk and to guarantee a high-quality SALTRACE data set of multiwavelength depolarization ratio profiles, we decided to run POLIS and BERTHA side by side during the entire SALTRACE-1 campaign at CIMH.

The full overlap of the laser beams of POLIS with the RFOV is at about 200 to 250 m above ground (Groß et al., 2016), so well within the marine boundary layer (MBL) and below the lofted Saharan air layer (SAL). The range resolution of the raw data is 3.75 m, the temporal resolution 5–10 s depending on atmospheric conditions. The repetition rate of the frequency doubled and tripled Nd:YAG laser is 10 Hz with a pulse energy of 50 mJ at 355 nm and 27 mJ at 532 nm (see Table 1 for more details).

## 2.5 CALIOP

Spaceborne lidar observations of the 532 nm particle linear depolarization ratio are used for comparison with the 532 nm particle depolarization ratios from the ground-based lidar observations during SAMUM-1, SAMUM-2, and SALTRACE. We analyzed all CALIOP observations (CALIPSO, 2016) for well-defined areas over Morocco ($26°–31°$N, $3°–8°$W), in the Cabo Verde region ($13°–18°$N, $21°–26°$W), and around Barbados ($10°–15°$N, $55°–60°$W), performed in June 2013 (11–13 overpasses), July 2013 (15–17 overpasses), June 2014 (13–16 overpasses), and July 2014 (11–14 overpasses). We checked all day and night overpasses (scenes) for the presence of dust and averaged all dusty signal profiles within the defined areas. In order to retrieve the dust depolarization ratio profiles for each overpass, only the observations characterized as dust from the CALIPSO subtype algorithm (Omar et al., 2009) are used. In these profiles, the particle linear depolarization ratio is recalculated from L2 perpendicular and total backscatter profiles, to improve the accuracy compared to the original CALIPSO L2-Version 3 product which has a known error (Tesche et al., 2013; Amiridis et al., 2013). Furthermore, several quality control procedures and filtering criteria are applied in the data set as described in (Marinou et al., 2017).

In the next step, we selected those height ranges (below 6 km height) of the monthly mean profiles in which the depolarization ratios were almost height independent, and computed the column-averaged 532 nm particle depolarization ratio for these specific height ranges. These column values are used for comparison in Sect. 4 and 5. The selected height ranges with almost height-independent depolarization ratios extended from 800–1000 m to 3700–5600 m (above sea level, Morocco), from 1500–2300 m to 4800–5400 m (Cabo Verde), and from 2500 to 3200–4200 m (Barbados). At lower heights, contamination with aerosol pollution and/or marine particles caused a significantly lower depolarization ratio. Therefore these lower heights were not considered in the dust-related depolarization data analysis.

## 2.6 AERONET photometers

Three sun photometers were run during the SALTRACE campaign at CIMH lidar station in 2013. Besides two CIMEL sun/sky photometers of AERONET (Aerosol Robotic Network) (Holben et al., 1998) from TROPOS and the University of Valladolid (see Barbados_SALTRACE, AERONET (2016)), an automatic sun/sky radiometer of the Meteorological Institute of the University of Munich measured the spectral aerosol optical thickness (AOT) and sky radiances (Toledano et al., 2009, 2011). The photometers covered a wavelength range from 340–1640 nm. The TROPOS photometer was operated from June 2013 to July 2014 (with an interruption from October 2013 to February 2014 caused by a damage of the sun photometer). Another photometer of AERONET is installed at Ragged Point (east coast of Barbados) in the vicinity of the Barbados Cloud Observatory (Stevens et al., 2016). The Ragged Point photometer performs measurements since 2007.

## 2.7 Radiosonde profiling

As during the SAMUM-1 and 2 campaigns we regularly performed radiosonde observations. The Vaisala RS92 radiosondes measuring height profiles of temperature, air pressure, relative humidity, wind speed and direction up to heights above 20 km were launched around local noon (15:00–16:00 UTC, 11:00–12:00 local time) and after sunset (23:00–24:00 UTC, 19:00–

20:00 local time). In total 133 radiosonde ascends were conducted at CIMH, 56, 35, and 42 during the SALTRACE-1, 2, and 3 campaigns, respectively.

## 3 SALTRACE case studies

Three case studies are presented to discuss the quality and accuracy of the spectrally-resolved depolarization-ratio observations with BERTHA. The first case study from summer 2013 offers the opportunity of direct comparisons with the Munich lidar system POLIS measured at the same field site in the framework of the SALTRACE-1 campaign (Groß et al., 2015). In summer 2014 only the lidar system BERTHA measured the transported Saharan dust. Two cases are shown, the first with a cirrus cloud, there the depolarization ratio is known, and the second, there the same dust reached one week later the North American continent and was measured by the HSRL-2 (Burton et al., 2015).

### 3.1 Case study I: Comparison of POLIS and BERTHA observations (11 July 2013)

A strong and long-lasting dust outbreak occurred from 9–13 July 2013. Figure 2 shows the BERTHA observations of the lofted SAL in the evening of 10 July 2013 (19:15–20:45 local time). The African air mass crossed Barbados with 15–20 m s$^{-1}$ wind speed from east to west according to the radiosonde profiles (upper left panel in Fig. 2). The relative humidity ranged from 30–50% in the dust layer between 1.75 and 4.6 km height (lower left panel in Fig. 2). The moist marine aerosol layer (MAL) indicated by high relative humidity around 80–90% reached to 1.75 km height on this evening. The marine boundary layer (MBL) is the convective part of the MAL and is often topped with trade wind cumuli. The MAL extends up to the base of the SAL which coincides with the trade wind inversion zone. Downward mixing of dusty air into upper part of the marine aerosol layer is visible in the lower right panel in Fig. 2 (green colors between 0.5 km and 1.5 km above sea level). This layer is also called intermediate layer (Jung et al., 2013), due to its location between the convective boundary layer and the SAL. The heterogeneous structures in the 532 nm volume depolarization height-time display in Fig. 2 below 1.5 km height are caused by island effects. Differences in orography and heat release over land and ocean surfaces disturb the air mass flow in the lowest part of the atmosphere (Jähn et al., 2016). Such vertical mixing features were not observed during the SALTRACE shipborne lidar observations over the open Atlantic in May 2013 (Kanitz et al., 2013; Rittmeister et al., 2017). The backward trajectories in Fig. 3 at 3000 m height indicate dust uptake over desert areas of Northwest Africa so that contamination with anthropogenic pollution was probably low. The dusty air masses traveled 5–7 days across the Atlantic to Barbados.

Figure 4 presents the particle optical properties obtained with the conventional Raman lidar technique (Ansmann et al., 1992). Typical features of Saharan dust were observed (Mattis et al., 2002; Papayannis et al., 2005; Tesche et al., 2011a; Preißler et al., 2011; Veselovskii et al., 2016; Hofer et al., 2017). The backscatter and extinction coefficients at 355 and 532 nm are similar and the dust 1064 nm backscatter coefficient is significantly lower than the respective 532 nm backscatter coefficient in the SAL. The observed Saharan dust lidar ratios accumulate in the 50–60 sr range at 355 and 532 nm. Below 1.75 km height a mixture of marine particles and dust particles prevailed so that the lidar ratio decreased. For pure marine conditions, the lidar ratio would be close to 15–25 sr (Flamant et al., 1998; Groß et al., 2011b; Burton et al., 2012; Dawson et al., 2015;

Rittmeister et al., 2017; Haarig et al., 2017). The SAL AOT was about 0.3 at 355 and 532 nm on this day. Figure 5 shows the particle linear depolarization-ratio profiles obtained with POLIS and BERTHA on 11 July 2013, 00:00–00:45 UTC. The POLIS backscatter coefficients in the left panels of Fig. 5 are computed by applying the Klett method with a dust lidar ratio of 55 sr within the SAL and 30–40 sr below the SAL (Groß et al., 2015). The Raman lidar method is used in the computation of the backscatter profiles from the BERTHA observations.

As can be seen, very good agreement is obtained regarding the volume linear depolarization ratio at 355 and 532 nm. In the computation of the particle depolarization ratio, the particle backscatter coefficients are required and cause further uncertainty. This impact is most sensitive at 355 nm. The apparent noise in the 355 nm particle depolarization ratio profiles is caused by the used backscatter coefficients. At 355 and 532 nm the SAL mean particle depolarization ratio can be compared between BERTHA (0.25 and 0.28 at 355 and 532 nm respectively) and POLIS (0.26 and 0.27). Both systems agree very well. Only BERTHA measured at 1064 nm, and derived a SAL mean particle depolarization ratio of 0.22. Below the SAL, all particle depolarization ratios decrease. The down-mixed dust prevent a decrease towards pure marine values of 0.02–0.03 (Groß et al., 2011b). In the right panel of Fig. 5, profiles of the ratio of particle depolarization ratios (355 nm over 532 nm in blue, 1064 nm over 532 nm in red) are shown. The mean values are 0.81 (1064/532 nm) and 0.88 (355/532 nm). The observed height-independence of the particle depolarization ratios at all three wavelengths and of the less noisy ratio of the 1064-to-532 nm depolarization ratios implies a vertically homogeneous dust size-shape characteristics. An impact of gravitational settling leading to a decrease of coarse-mode dust concentration in the SAL top region after 5–10 days of travel, which would show up in a significant change in the spectral slope of the depolarization ratio (especially at 1064 nm), is not visible. This corroborates the hypothesis proposed by Gasteiger et al. (2017) that heating of the dust particles and turbulent mixing of the SAL air masses during daytime hours may widely reduces coarse-mode dust removal by gravitational settling of particles.

## 3.2 Case study II: Measurements of the dust layer and a cirrus (20 June 2014)

During SALTRACE-3 in 2014, POLIS was not available. Cirrus depolarization measurements were used to check the quality of triple-wavelength depolarization observations over time. Ice crystals are very large compared to the laser wavelengths so that the spectral dependence of backscattering, extinction, and depolarization properties is rather weak.

Figure 6 presents a cirrus measurement performed on 20-21 June 2014. The cirrus layer between 12 and 14 km height was optically thin with an AOT of 0.1. The wavelength-independent backscatter coefficients of up to 3.5 $Mm^{-1}$ $sr^{-1}$ at cirrus center indicate peak particle extinction values of 100-120 $Mm^{-1}$. The extinction values are obtained by applying a multiple-scattering-corrected cirrus lidar ratio of 30–35 sr to the cirrus backscatter coefficients (Haarig et al., 2016b). The extinction measured with the Raman channels would need a too large smoothing length in the thin cirrus.

As shown in the right panel of Fig. 6, depolarization ratios within the cirrus (above 12.2 km height) for 532 and 1064 nm are almost equal up to cloud top. The 1064 nm particle linear depolarization ratio is close to 0.5 and height-independent from 12.2–13.3 km height. The noisy 355 nm particle depolarization ratio is less trustworthy, but close to the 532 nm and 1064 nm depolarization ratios at least in the cirrus backscatter center from 12.4–12.8 km height. This consistent cirrus measurement

of wavelength-independent cirrus backscatter and depolarization corroborate that BERTHA was performing well and that our dust observations below 5 km height are trustworthy.

Figure 7 shows the aerosol layers in the lower troposphere on this cirrus day. A 3 km thick SAL was present above the marine aerosol layer. Relative humidities of 40–50% in the SAL were comparably high and suggest some mixing with moist marine air.

The HYSPLIT backward trajectories in Fig. 8 corroborate this hypothesis. 4 days before arriving at Barbados, the air masses at 2.5 km height had the chance of vertical mixing with marine particles or African pollution. Only the uppermost dust layer (3–4 km height over Barbados, trajectories not shown) seems to contain pure dust. The air masses arriving at 3.5 km height were above 6 km over desert areas in western Africa.

Figure 9 presents the particle optical properties derived from the BERTHA observations. A steady and almost monotonic decrease of the backscatter coefficients with height was found. The extinction coefficients at 355 and 532 nm are again very close and the SAL AOT was about 0.25. The lidar ratios range from 40–50 sr in the layer from 1–3 km height, which probably contained some marine and anthropogenic haze particles, and was higher with values of 50–60 sr in the uppermost pure dust layer (3–4 km height).

The height profiles of the 532 and 1064 nm particle depolarization ratio show slightly different profile shapes. The 1064 nm depolarization ratio decreases slightly with height whereas the 532 nm increases with height. Since the 1064 nm depolarization ratio is very sensitive to coarse-mode particles, the decrease of the 1064 nm depolarization ratio (and of the ratio of 1064-to-532 nm depolarization ratios) may be related to a decreasing coarse-mode mean radius of the particles with height. The 532 nm depolarization ratio, on the other hand, seems to be relatively insensitive against small changes in the coarse-mode size distribution (Mamouri and Ansmann, 2017). The 532 nm depolarization ratio for coarse mode particles is always in the range of 0.35-0.4. The slightly lower 532 nm depolarization values around 2 km height may again indicate a different history of the dust transport. The more fine-mode sensitive depolarization ratios (355 and 532 nm) and the respective ratio of 355-to-532 nm depolarization ratios show an almost height-invariant behavior when ignoring the noise in the 355 nm depolarization ratio profile.

### 3.3 Case study III: Dust transport from Africa to Missouri over 12000 km (6 July 2014)

A unique case was observed during SALTRACE-3 on 6 July 2014. A 3 km deep dust layer crossed Barbados and traveled westward towards the United States (see backward trajectories in Fig. 10). Coincidentally, this aged dust layer was observed with an airborne triple-wavelength polarization lidar (high spectral resolution lidar HSRL-2) one week later (Burton et al., 2015). We use this unexpected opportunity to compare the triple-wavelength depolarization observations of aged dust after long-range transport over 6000 km (Barbados) and 12000 km (Missouri, Midwestern USA). Rather low relative humidities around 20% were measured with radiosonde in the lofted SAL (see lower left panel in Fig. 11) suggesting almost no interference with cloud formation and associated upward mixing of marine air into the lower part of the SAL. High wind speeds around 18 m s$^{-1}$ prevailed above 2.5 km height over Barbados. The 12.5-day backward trajectories indicate that the dust observed over Barbados at 2.5 and 3.5 km height descended towards 1.6 and 2.4 km height over Missouri, after crossing Yucatan

(Mexico), Texas, and Oklahoma. After leaving the Africa continent, the dust layers arrived after 5 and 12 days over Barbados and Missouri, respectively.

Fig. 12 shows the aerosol optical properties derived from the BERTHA measurements. The triple-wavelength particle linear depolarization observations performed over Missouri are added. The backscatter, extinction, and lidar ratio profiles again show typical dust optical properties. The SAL AOT was about 0.3 over Barbados on this day. The backscatter and extinction coefficients were roughly 25-30% lower in the lofted dust layer over Missouri compared to the values measured over Barbados one week before.

An excellent agreement between the two lidar data sets of depolarization ratio profiles was found. When comparing the values in the layers from 2.5–4.0 km height over Barbados with the values in the layer from 1.5–2.0 km height over Missouri, both lidars found similar depolarization ratios. At 532 nm (0.28 (BERTHA) and 0.30 (HSRL-2)) and 1064 nm (0.26 (BERTHA) and 0.27-0.28 (HSRL-2)), the depolarization ratio over Barbados is slightly lower. The 355 nm depolarization ratio measured with BERTHA with a larger systematic uncertainty is around 0.24 and therefore higher than over Missouri (0.21 (HSRL-2)). But spectral behavior of the 1064 nm to the 532 nm depolarization ratio was the same over Barbados and Missouri (0.91). The 355/532 ratio of depolarization ratios decreased between Barbados and the US.

Note also, the higher depolarization ratios in the planetary boundary layer over Missouri. Probably new soil dust particles were injected into the continental planetary boundary layer and then mixed upward into the lofted aged SAL over the United States. This entrainment of fresh coarse dust influences especially the 1064 nm depolarization ratio. Fine-mode pollution is released as well over the United States, and upward mixing and entrainment of pollution aerosol into the SAL affects most sensitively the 355 nm depolarization ratio. Nevertheless, the agreement is surprisingly good. Even after 7 days of travel from Barbados to the Midwestern United States, the Saharan aerosol widely preserved its dust characteristics. No significant height-dependence of the optical properties (lidar ratio, depolarization ratio, ratio of depolarization ratios) is visible in the main parts of the layer over both sites.

## 4  SALTRACE statistical overview

Figure 13 provides an overview of all triple-wavelength depolarization-ratio observations with BERTHA during the summer SALTRACE campaigns. SAL mean values of 8 evening sessions (SALTRACE-1, summer 2013) and 13 evening session (SALTRACE-3, summer 2014) are shown. The respective mean values derived from the BERTHA observations in June 2013 and 2014 and July 2013 and 2014 are shown as horizontal lines. In addition, monthly means of the SAL column dust depolarization ratio obtained from spaceborne lidar CALIOP observations at 532 nm in the Barbados region are presented.

The number of evening observations with a complete set of depolarization ratios at all three wavelengths is comparably low for SALTRACE-1 (2013) because of many days with rainy and cloudy weather, days without dust, and also due to problems with one of the lasers. In 2014, 13 evening data sets of triple-wavelength depolarization ratio profiles out of a total of 21 possible evening lidar sessions could be used for the statistics in Fig. 13. Less days with closed cloud decks and rain periods hampered observations in 2014. The five-week SALTRACE-1 field phase was embedded in a typical tropical wet season. Short-term dust

episodes were frequently interrupted by rainy weather. Radiosonde profiles often showed different wind directions from south to northeast within the 3–5 km thick Saharan dust layers until 9–10 July 2013. Cloud formation and activation of dust particles to serve as cloud condensation nuclei, rain and corresponding washout probably significantly influenced the dust characteristics during the long-range transport across the Atlantic. A pronounced, well-defined dust outbreak lasting over several days was

5 observed from 9–13 July only during the SALTRACE-1 campaign in 2013. Most of the July 2013 observations in Fig. 13 were taken during this final SALTRACE-1 dust outbreak. In contrast, more homogeneous, vertically well-structured dust outbreaks were observed in June and July 2014. The summer of 2014 was an extraordinary dry season in the Barbados area and over the tropical Atlantic. Cloud processing and washout by rain was strongly suppressed in the summer of 2014. Continuous and almost undisturbed dust transport from Africa towards the Caribbean occurred.

The fluctuations in the individual depolarization ratio values in Fig. 13 partly reflect the impact of cloudy and rainy weather. The mean values (horizontal lines) for July 2013 (with the well-defined dust outbreak), June and July 2014 differ significantly from the ones for June 2013 for the wavelengths of 532 and 1064 nm. On average, the 532 and 1064 nm particle depolarization ratios accumulated in the 0.28–0.30 and 0.22–0.26 range, respectively, during the more dry and less cloud and rain-affected periods. A similar contrast (wet 2013 vs dry 2014 months) is observed with CALIOP. The CALIOP monthly means include

SAL column values around Barbados (10°–15° N, 55°–60° W). The large standard deviation bars of the CALIOP monthly means in Fig. 13 mainly indicate the atmospheric variability within the defined areas given in Sect. 2.5. Good agreement between the BERTHA and the CALIOP measurements is found.

A considerable part of the scatter in the BERTHA data is however caused by retrieval uncertainties (see the systematic uncertainty bars in Fig. 13). These uncertainties are rather large for 355 nm. The error bars of the individual measurements

only show the variability (standard deviation, SD) around the mean values within the observed individual SAL height range from base to top. The systematic errors of the 1064 nm particle depolarization ratio are comparably small. Thus the fluctuations of the 1064 nm depolarization ratio indicate the changes in the dust microphysical characteristics from day to day. The large-particle fraction is expected to vary with time as a function of varying dust removal strength due to different travel conditions across the Atlantic and different weather conditions. This especially influences the 1064 nm SAL mean depolarization ratio

according the discussion below. The depolarization ratio measurements at 1064 nm will improve the retrieval of microphysical properties according to Gasteiger and Freudenthaler (2014).

Table 2 provides the overall SALTRACE (summers of 2013 and 2014) mean depolarization ratio values for all three wavelengths, the associated SD values (showing the day-to-day variability of the SAL), and typical systematic retrieval errors. Typical systematic errors are obtained from averaging of the respective uncertainties of the 21 evening observational cases.

The SALTRACE mean depolarization ratios are 0.25 (355 nm), 0.28 (532 nm), and 0.23 (1064 nm).

We found good agreement between the two depolarization data sets collected with POLIS (Groß et al., 2015) and BERTHA for July 2013. On average, particle depolarization ratios were 0.28 (POLIS) and 0.29 (BERTHA) at 532 nm and 0.27 (POLIS) and 0.26 (BERTHA) at 355 nm. For June 2013, the results at 532 nm differ significantly. On average, we observed with BERTHA June 2013 means of 0.26 (532 nm) and 0.26 (355 nm) based on four individual evening measurements. The POLIS

June 2013 mean values (based on three evening measurement sessions) were higher at 532 nm (0.29) and equal at 355 nm

(0.26). The large systematic uncertainties in the BERTHA depolarization ratios are probably mainly responsible for the observed differences as well as the low number of observations which is too low for drawing solid conclusions on the quality of the measurements.

## 5  Discussion

Figure 14 provides an overview of the entire SAMUM-SALTRACE data set of depolarization ratios collected from 2006 to 2014 in southeastern Morocco (SAMUM-1), at Praia, Cabo Verde (SAMUM-2), and on Barbados (SALTRACE). There is almost no change of the mean 355 nm and 532 nm particle depolarization ratio with distance from the dust source when combining the BERTHA (Barbados) and POLIS/MULIS (Cabo Verde, Morroco) data sets. The POLIS/MULIS data sets indicate a slow decrease of the mean 532 nm depolarization value from 0.31 (Morocco) over 0.30 (Cabo Verde) toward 0.28 (Barbados), but the difference is not significant. The available 1064 nm mean values indicate a significant decrease from a Morocco mean value of 0.27 to a Barbados mean value of 0.23. We speculate that a large fraction of large dust particles causing depolarization ratios of 0.40 is present over areas close to the Sahara but that these large particles are widely removed before reaching Barbados. The other two wavelengths (355 and 532 nm) are more sensitive to fine-mode dust (accumulation-mode particles with diameters <1 $\mu$m), for which the removal by gravitational settling is less efficient. Especially, the 355 nm SAL values, widely controlled by fine-mode dust particle, show a rather robust behavior. No trend in the 355 nm depolarization values is observed in the SAMUM-1, 2 and SALTRACE data.

Figure 14 includes CALIOP depolarization measurements performed in the Morocco, Cabo Verde, and Barbados region in June and July of 2013 and 2014. The overall four-month mean values (plus SD) are shown. The CALIOP 532 nm mean values (and SD) are 0.31±0.07 (Morocco), 0.30±0.05 (Cabo Verde), and 0.30±0.06 (Barbados). Good agreement between the ground-based BERTHA and spaceborne CALIOP observations is obtained, and no significant trend in the 532 nm particle depolarization ratio found. Veselovskii et al. (2016) reported an average 532 nm particle linear depolarization ratios of 0.30±0.045 for the SHADOW campaign in Senegal during dust outbreaks in March-April 2015.

The found spectral slope of the depolarization ratio, shown in Fig. 15, with the maximum at 532 nm and lower values at 355 and 1064 nm reflects the different influence of the fine-mode and coarse-mode dust fractions on the overall (fine + coarse) particle depolarization ratio at the three wavelengths. The 355 nm dust particle depolarization ratio is strongly influenced by fine-mode dust (up to 50–60% fine mode fraction (FMF) according to AERONET observations). Fine-mode dust causes a depolarization ratio around 0.20 at 355 nm. The comparably weak influence of coarse-mode dust (causing depolarization ratios >0.30) leads to an overall (fine + coarse-mode) particle linear depolarization ratio of 0.25±0.03. The 532 nm dust depolarization ratio is still sensitively influenced by fine-mode dust (FMF 10–30% contribution, causing a fine-mode depolarization ratio around 0.15) but also by the coarse-mode dust particle fraction (leading to a depolarization ratio of 0.35–0.40). The overall effect of fine-mode and coarse-mode depolarization then leads to the observed 532 nm depolarization ratios around 0.30±0.03. In contrast, the 1064 nm dust particle depolarization ratio is caused to about 95% by coarse-mode dust particles (FMF 5%) according to AERONET observations, and seems to be between 0.20–0.28 for dust after long-range

transport. The influence of the fine and coarse-mode dust fractions on the particle linear depolarization ratio is discussed in detail by Mamouri and Ansmann (2017).

Simulation studies of Kemppinen et al. (2015a) can be used to interpret our depolarization observations. These simulations are based on realistic dust particle shapes, measured during SAMUM-1 in Morocco (Lindqvist et al., 2014). For the so-called

dolomite shape type, the simulations yield 1064 nm and 532 nm coarse-mode particle linear depolarization ratios of 0.20–0.25 and 0.35–0.40, respectively, provided dust particles with diameters around 2 $\mu$m dominate backscattering of laser photons. These depolarization values are in good agreement with the 1064 nm lidar observations as well as with 532 nm depolarization studies when taking a dust fine-mode fraction of 20% (as indicated by AERONET sun photometer observations), a fine-mode depolarization ratio of 0.15, and a measured 532 nm depolarization ratio of 0.30 into account, as explained in de-

tail by Mamouri and Ansmann (2017). It is interesting to note in this context that the AERONET photometer observations during dust outbreaks over Barbados in 2013 and 2014 show that the coarse-mode effective diameter accumulates around $3\pm0.4\mu$m which indicates that most coarse-mode dust particles after long-range transport have diameters in the 1-3 $\mu$m size range. Veselovskii et al. (2016) retrieved overall (fine-mode + coarse-mode) effective diameters of 2–2.5 $\mu$m from the multi-wavelength lidar measurements in the dust plumes over Senegal during the SHADOW campaign.

This consistency between the lidar observations, photometer retrievals and simulation studies is promising and suggest that our triple-wavelength polarization lidar observations are very useful for next steps in dust simulation studies with the goal to develop an appropriate dust size-shape parametrization scheme for atmospheric weather and climate models. However, the consistency created here is not more than an hypothesis. Much more laboratory and model studies together with our complex field observations are required to improve stepwise our knowledge on the complex relationship between fundamental dust

properties and related optical effects.

Figure 15 provides a comparison of the observations with triple-wavelength polarization lidars (BERTHA, HSRL-2) and a preliminary modeling result of the depolarization ratios described by Gasteiger et al. (2017) in their hypothesis with vertical mixing during the day. A mix of irregularly shaped (non-spheroidal) particles and small spherical ammonium sulfate particles is assumed in the simulation. The mineralogical variability is mimicked by mixing of absorbing and non-absorbing particles.

The spectral trend observed with the lidars is also visible in the simulation. The deviation of the modeled spectral slope of the depolarization ratio from the observed wavelength dependence may be related to the shape parametrization of the rough estimate of the dust size distribution. Fig. 15 may be regarded as the starting point of a comprehensive modeling effort. A dense observational data on depolarization ratios together with airborne in situ observations of size distributions and the mineralogical/chemical composition of the dust particles in the Barbados region (Weinzierl et al., 2017) is now available for

in-depth simulation studies of dust optical properties.

## 6   Conclusions

Triple-wavelength polarization lidar measurements in long-range transported Saharan dust layers were performed at Barbados, 5000–8000 km west of the Saharan dust sources, in the framework of three SALTRACE campaigns, each lasting over several

weeks. High quality was achieved by comparing the BERTHA observations with depolarization ratio profiles measured with a reference system and by using cirrus layers in which the spectral dependence of the particle depolarization ratio vanishes. A unique case of long-range transported dust over more than 12000 km was presented and indicated widely unchanged Saharan dust optical properties even after a travel time of two weeks since the emission. On average, the particle linear depolarization ratios for aged Saharan dust were found to be 0.252±0.030 at 355 nm, 0.280±0.020 at 532 nm, and 0.225±0.022 at 1064 nm (mean ± standard deviation). According to published simulation studies we conclude that most of the coarse-mode dust particles have sizes around 2 $\mu$m in diameter after one week of travel. By comparing the SALTRACE results to the SAMUM-1 and SAMUM-2 results, again, only minor changes in the dust depolarization characteristics were observed on the way from the Saharan dust sources towards the Caribbean. Only the 1064 nm depolarization ratio mean value decreased significantly from Morocco towards the Caribbean.

A long-term data set of the particle linear depolarization ratio of mineral dust measured simultaneously at 355, 532, and 1064 nm is now available. In addition, dense 355 and 532 nm lidar ratio data sets are available (Groß et al., 2011a, 2015; Tesche et al., 2011a). Furthermore, airborne in situ observations of the dust particle size distribution and chemical composition in the SAL are available (Weinzierl et al., 2017). This is an excellent basis for comprehensive simulation efforts to develop realistic dust shape models and parametrization schemes which link the dust size distribution, composition, and shape characteristics with the resulting optical and radiative properties of mineral dust particles.

The available coherent multiwavelength data sets on linear depolarization ratios and lidar ratios (from the source region to remote areas of long-range transport) support present and upcoming spaceborne lidar missions (CALIPSO and EarthCARE missions) and the development of new space lidar mission concepts (based on multiwavelength polarization/HSRL lidar missions). They can further be used to harmonize existing and future depolarization data sets collected at different lidar wavelengths. Furthermore, existing dust retrieval schemes such as the technique presented in this SALTRACE special issue by Mamouri and Ansmann (2017) can be checked and improved based on the available complex depolarization ratio data sets.

Together with the observations of the 355 and 532 nm depolarization ratios with POLIS (EARLINET reference system) a high quality data set on depolarization ratios at 355, 532, and 1064 nm for Saharan dust after long range transport is now available for the first time. Comparison with another triple-wavelength depolarization ratio data set indicates that the Barbados data is very trustworthy. CALIOP depolarization ratios collected over the SAMUM-1, SAMUM-2, and the SALTRACE field sites are in very good agreement with the findings of the ground-based lidars. Altogether, a significant dust aging effect triggering a significant change of the dust depolarization ratio from regions close to the source to areas more than 5000 km downwind is not visible in the observations. Discrepancies between the modeled and the observed depolarization ratios are not surprising when keeping in consideration that the shape characteristics of the irregularly shaped dust particles is not well known, and a realistic shape model is not existing.

As an outlook, we are presently testing to measure besides dust depolarization ratios at three wavelengths also the dust extinction coefficient at these three wavelengths. First test observations were promising (Haarig et al., 2016b).

## 7 Data availability

HYSPLIT backward trajectories are calculated via the available simulation tools (HYSPLIT, 2016). AERONET sun photometer AOT data are downloaded from the AERONET web page (AERONET, 2016). SALTRACE BERTHA lidar data are available at TROPOS. CALIOP signal profile data are made available by the CALIPSO science team (CALIPSO, 2016). We used level 2 version 3.30 CALIPSO Aerosol profile data.

**Appendix A: Polarization channels and their calibration in the lidar system BERTHA**

The appendix A explains shortly the state of the art concept of polarization lidar according to Freudenthaler (2016) and discusses some special features of the BERTHA lidar system. In the second part, the BERTHA lidar system will be characterized in detail and the systematic errors will be estimated.

## A1 Calculation of the depolarization

In a state-of-the-art approach the components of the lidar system are described by Mueller matrices using the Mueller-Stokes formalism (Freudenthaler, 2016):

$$\boldsymbol{I}_{\mathrm{S}} = \eta_{\mathrm{S}}\mathbf{M}_{\mathrm{S}}(D_{\mathrm{S}},\nu_{\mathrm{S}},\Delta_{\mathrm{S}})\mathbf{C}(\Psi,\epsilon)\mathbf{M}_0(D_0,\gamma,\Delta_0,a_0)\mathbf{F}(a)\mathbf{M}_{\mathrm{E}}(D_{\mathrm{E}},\beta,\Delta_{\mathrm{E}},a_{\mathrm{BE}})\boldsymbol{I}_{\mathrm{L}}(\alpha) \tag{A1}$$

The different quantities have to be known or at least good estimates and their uncertainties are needed. They will be described in the following text and determined for each wavelength separately in the next section. An overview is given in Table 3.

All the rotational misalignment around the optical axis (rotational misalignment in the following) is with respect to the plane of polarization of the polarization filter in front of the PMT belonging to the cross polarized channel of the corresponding wavelength. The emitted laser light $\boldsymbol{I}_{\mathrm{L}}$ is has a rotational misalignment $\alpha$.

The emitting optics $\mathbf{M}_{\mathrm{E}}$ (beam expander and steering mirrors) have a diattenuation parameter (called diattenuation in the following) $D_{\mathrm{E}}$, a retardation $\Delta_{\mathrm{E}}$ and a misalignment $\beta$. Furthermore the beam expander may cause depolarization due to inhomogeneities over its surface and birefringence of the calcium fluoride lens. The degree of linear polarization after the beam expander is given by $a_{\mathrm{BE}}$.

$\mathbf{F}$ represents the scattering process in the atmosphere with an atmospheric polarization parameter $a$ corresponding to the
20 atmospheric volume linear depolarization ratio $\delta$:

$$a = \frac{1-\delta}{1+\delta} \tag{A2}$$

which is the unknown quantity, that will be derived by polarization lidar measurements.

The receiving optics $\mathbf{M}_0$, in the case of BERTHA only the telescope and a 90° mirror, are characterized by the diattenuation $D_0$, the retardation $\Delta_0$ and the misalignment $\gamma$.

A linear polarizer is used for the calibration C, the position $\Psi$ is +45° or -45° and $\epsilon$ is the deviation of this ideal position, but not affecting the exact 90° angle between the two calibration positions (±45°). The exact difference is reached by a highly accurate ($10^{-4}$ degrees) step motor (8SMC1 from Standa Ltd., Lithuania). The calibrator is only used during the calibration measurements. For the regular measurements, it is taken out of the beam and therefore a possible misalignment is not affecting the measurements.

The detection units for the total channel (index S=T) and for the cross polarized channel (index S=R), $\mathbf{M}_{\mathrm{T}}$ and $\mathbf{M}_{\mathrm{R}}$, include a diattenuation, which would be ideally $D_{\mathrm{T}}=0$ and $D_{\mathrm{R}}=-1$. In addition to Freudenthaler (2016) a rotational misalignment $\nu_{\mathrm{S}}$

and a retardation $\Delta_S$ has to be included in the formulas for the BERTHA lidar system. The calibrator is located behind the telescope and before the beam separation unit (see Fig. 1). The retardation $\Delta_S$ has no effect on the measured intensity and need no further consideration. For the total channel there is no rotational misalignment, only $\nu_S$ for the cross polarized channel has to be taken into account. The gain ratio of the PMT and its neutral density filters is represented by $\eta_T$ and $\eta_R$, respectively. A calibration measurement for every change in the neutral density filters was performed.

Freudenthaler (2016) presents a solution to the matrix equation (eq. A1) introducing the parameters $G_S$ and $H_S$ as system constants. They are a simplification in the notation of the solution and depend on all previous mentioned system parameters (Tab. 3). The remaining unknown is the atmospheric polarization parameter $a$. The intensity (first component of the Stokes vector $\boldsymbol{I}_S$) is given by:

$$I_S = \eta_S T_S T_0 F_{11} T_E I_L \left( G_S + a H_S \right) \tag{A3}$$

with the laser intensity $I_L$, the transmittance of the emitting optics $T_E$, of the receiving optics $T_0$, of the detection unit $T_S$, the backscatter coefficient $F_{11}$ and the gain of the detector $\eta_S$. Taking the ratio of two signals, only $T_S$ and $\eta_S$ are remaining. The volume linear depolarization ratio $\delta_v$ can be determined:

$$\delta_v = \frac{\delta^*(G_T + H_T) - (G_R + H_R)}{(G_R - H_R) - \delta^*(G_T - H_T)} \tag{A4}$$

with the calibrated signal ratio $\delta^*$:

$$\delta^* = \frac{1}{\eta} \frac{I_R}{I_T} \tag{A5}$$

with calibration factor $\eta$, determined by the $\Delta 90°$-calibration (Freudenthaler et al., 2009). The measured calibration factor $\eta^*$ has to be corrected for any of the above mentioned rotational misalignment. The correction constant K is used to obtain the calibration factor $\eta$ (Freudenthaler, 2016):

$$\eta = \frac{\eta^*}{K} \tag{A6}$$

The five parameters $G_T$, $H_T$, $G_R$, $H_R$ and K (system constants) are calculated from the system characteristics (Tab. 3) for each wavelength.

## A2    System characterization

The results of the system characterization are shown in Table 3. Similar characterizations of different EARLINET lidars have been reported by Bravo-Aranda et al. (2016). The results are shown for the conditions during SALTRACE-3 as it has been

characterized when the system was back in Leipzig. Some characterization and optimization efforts have already been done at the field site on Barbados. One year in the tropical region and the transport might influence the system performance.

   The degree of linear polarization of the laser is high (>95% according to manufacturer's specification) and it is additionally cleaned by a Glen-Taylor polarizer, whose misalignment with respect to the optical table is characterized by the angle $\alpha$ and measured with the help of an additional polarization filter with the uncertainty of 0.2°.

The beam expander is the major source of uncertainty in the emitter optics, as it has to deal with the three wavelengths simultaneously and the high power of the laser. The used CaFl lens appears to be birefringent. In order to check this issue, the beam expander was tested in the EARLINET lidar calibration center in Bucharest (www.lical.inoe.ro). The Mueller matrix elements have been measured with transmission ellipsometry at 25 points over the surface of the beam expander. Inhomogeneities of the Mueller matrix elements in the external regions of the beam expander have been found, especially in the UV. As a first approximation a degree of linear polarization of $0.97 \pm 0.02$ in the UV after the beam expander is assumed. In the visible and near infrared the degree of linear polarization is almost ideal (0.99 to 1.00). The influence of the other components (mainly steering mirrors) of the emitter optics is neglected so far, although the diattenuation of the emitter optics is mainly induced by these steering mirrors. From the transmission ellipsometry of the beam expander a diattenuation ($D_E$) smaller than 0.05 is derived. Not all elements of the Mueller matrix of the beam expander could be measured. Therefore the retardation $\Delta_E$ of the emitter optics is not known, and the maximum uncertainty of $\pm 180°$ is assumed. The same assumption has been made for other EARLINET lidars by Bravo-Aranda et al. (2016). An appropriate assumption for the rotational misalignment $\beta$ is $0° \pm 1.0°$. The receiver diattenuation ($D_0, D_T, D_R$) was characterized using an additional light source with known state of polarization ($0°$ to $360°$ in steps of $1°$) similar to Mattis et al. (2009). A second calibration measurement was performed with the internal linear polarizer after the telescope to separate the influence of $D_0$ from $D_T$ and $D_R$. The two calibrations indicated a depolarization of the $90°$-mirror after the secondary mirror (see Fig. 1). The polarization parameter $a_0$ of this mirror has a significant influence at 532 nm only ($a_0$=0.958). For the UV ($a_0$>0.994) and the near infrared ($a_0$>0.994) it can be attributed to uncertainties in the measurement ($\pm 0.005$) and its influence is thus neglected, but considered in the error estimation. The rotational misalignment $\gamma$ of the receiving optics is assumed to be $0° \pm 0.2°$. It is a fixed, well-aligned system. The retardation of the receiver could not be measured, which lead to the maximum uncertainty of $\pm 180°$. The rotational misalignment ($\nu_R$) in the cross polarized channel after the calibrator is assumed to be $0° \pm 0.5°$.

The deviation of the ideal calibration position $\epsilon$ can be estimated according to Freudenthaler (2016) and is only present during calibration measurements, while at the normal measurements the linear polarizer is taken out of optical path. It varies between $-3.1°$ and $1.8°$ at 355 nm, $-2.4°$ and $1.5°$ at 532 nm and $-1.1°$ and $2.4°$ at 1064 nm. The parameter K corrects the calibration constant for the misalignment $\epsilon$. For each calibration the parameter K was calculated using the estimated $\epsilon$. The corrections are small as $|\epsilon|$<5°.

K is mainly influenced by the contrast ratio (=1/extinction ratio) of the linear polarizer used for the calibration. The manufacturer CODIXX specifies the contrast ratio to be 1:10,000 (at 355 nm), 1:60,000 (at 532 nm) and 1:90,000 (at 1064 nm).

The 12 parameters listed in Table 3 introduce an uncertainty of the volume linear depolarization ratio. It is too complex to perform classical error propagation. Therefore a Monte-Carlo simulation is used as proposed by Bravo-Aranda et al. (2016). This is actually a complete search over the multi-dimensional uncertainty space, there the 12 parameters are varied within their error margins. Uniform distributions are used as input. The rotational misalignment ($\alpha, \beta, \gamma, \nu_R$) has been simulated with three input values $x_i - \Delta x_i$, $x_i$, $x_i + \Delta x_i$. The diattenuation ($D_E, D_0, D_T, D_R$) has been simulated with five input values $x_i - \Delta x_i$, $x_i - 0.5\Delta x_i$, $x_i$, $x_i + 0.5\Delta x_i$, $x_i + \Delta x_i$ as it is a are more sensitive parameter (Bravo-Aranda et al., 2016). The unknown retardation ($\Delta_E$ and $\Delta_0$) is simulated using 5 values as well: $-180°$, $-90°$, $0°$, $90°$ and $180°$. The polarization parameter of the

90°-mirror in the telescope $a_0$ has been simulated with three values, for 355 and 1064 nm 1.000, 0.995 and 0.990 were used. The degree of linear polarization after the beam expander $a_{BE}$ has been simulated with three values for 532 nm and 1064 nm (1.000, 0.995 and 0.990) and 5 values for 355 between 0.95 and 0.99 in steps of 0.01. The variation of $\epsilon$, which does not influence the calculation of $G_S$ and $H_S$, has not been taken into account to reduce the number of variables by a factor of 3. In the Monte-Carlo simulation over 11 million combinations are used for 532 nm and 1064 nm. For 355 nm over 18 million

combinations are used as input. The original code, based on Freudenthaler (2016), without our modifications as mentioned above, can be found on https://bitbucket.org/iannis_b/ (last accessed Feb. 2017).

    The simulation has been performed for the theoretical Rayleigh values (0.0080, 0.0053, 0.0036), and typical values within the dust layer (0.08, 0.20, 0.23) for 355 nm, 532 nm and 1064 nm, respectively. For an input value (true value) the program simulates the calibrated signal ratio $\delta^*$ varying all lidar parameters within their error margins as described above. Then it cal-

culates the volume depolarization ratio $\delta_v$ according to equation A4. The frequency distributions of the solutions are shown in Fig. 16. The minimum and maximum value as well as the standard deviation from the corresponding Gaussian distribution is summarized in Table 4.

The standard deviation is the measure for the systematic error of the volume linear depolarization. It is slightly higher in the dust layer than in the aerosol-free Rayleigh background.

The systematic error for the particle linear depolarization ratio mainly includes the uncertainties of the volume depolarization ratio (Table 4) and the backscatter ratio (molecular backscatter coefficient divided by particle backscatter coefficient). For the particle backscatter coefficient an uncertainty of 5%, 10% and 15% for 355, 532 and 1064 nm, respectively, is assumed. Less Rayleigh signal towards higher wavelengths increases the uncertainty in the reference value for the calculation of the particle backscatter coefficient. The uncertainty of the molecular backscatter coefficient is <1% because it is calculated from the

temperature and pressure profiles of the radiosondes which were launched twice a day at the time of the measurements. The molecular depolarization $\delta_m$ is calculated from the transmission curves of the interference filters and depends on temperature (Behrendt and Nakamura, 2002). An error of 10% is assumed.

*Acknowledgements.* The perfect logistic support of CIMH during the SALTRACE preparation phase and intensive field phases in 2013

and 2014 is gratefully acknowledged. We are grateful to the AERONET team for performing high quality SALTRACE sun photometer calibrations and for providing high quality data products. We would like to express our gratitude to the CALIPSO science team for the careful CALIOP profile data analysis. We thank the HYSPLIT team for the possibility to compute backward trajectories. We are grateful to Laurentiu Baschir and the Lidar Calibration Center (Lical), Bucharest, Romania, for excellent service regarding the characterization of lidar optical elements. This activity is support by ACTRIS Research Infrastructure (EU H2020-R&I) under grant agreement no. 654169

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

**Table 1.** Main system parameters the BERTHA and POLIS lidar systems.

| Property | BERTHA | POLIS |
|---|---|---|
| Emitted wavelengths (nm) | 355, 532, 1064 | 355, 532 |
| Pulse energy (mJ) [*] | 120 + 800 + 1000 | 50 + 27 |
| Repetition rate (Hz) | 30 | 10 |
| Telescope | Cassegrain | Dall-Kirkham |
| Eff. telescope diameter (m) | 0.53 | 0.175 |
| Receiver field of view (mrad) | 0.8 | 2.5 |
| Detected wavelenghts (nm) | | |
| - elastic (total) | 355, 532, 1064 | |
| - co-polarized | | 355, 532 |
| - cross-polarized | 355, 532, 1064 | 355, 532 |
| - inelastic (vib. Raman) | 387, 407, 607 | 387, 607 |
| - further | $532_{HSRL}$, $532_{RR}$, | |
| Range resolution (m) | 7.5 | 3.75 |

HSRL - High Spectral Resolution Lidar; RR - Rotational Raman

* internal attenuation are not taken into account for BERTHA, but for POLIS

**Table 2.** SALTRACE mean particle linear depolarization ratio and corresponding standard deviation (SD), for all three wavelengths, observed within the Saharan dust layer with BERTHA in June and July 2013 and June and July 2014. Typical uncertainties in the retrieval of individual particle depolarization ratio values (systematic errors) are given for comparison. These uncertainties consider the volume depolarization retrieval uncertainties (see Appendix) and uncertainties in the backscatter coefficients required as input in addition.

| Wavelength | Mean | SD | Syst. error |
|---|---|---|---|
| 355 nm | 0.252 | 0.030 | 0.074 |
| 532 nm | 0.280 | 0.020 | 0.019 |
| 1064 nm | 0.225 | 0.022 | 0.008 |

**Table 3.** BERTHA parameters (values) used in the volume depolarization ratio retrieval and uncertainties. Each wavelength is treated separately. Rotational misalignment of the laser ($\alpha$), of the emitting optics ($\beta$), the receiving optics ($\gamma$) and the cross polarized receiver channel ($\nu_R$); diattenuation and retardation of the emitter optics($D_E$, $\Delta_E$) and the receiver optics ($D_0$, $\Delta_E$); the diattenuation of the total and cross-polarized receiver channel ($D_R$,$D_T$); the degree of linear polarization of the beam expander ($a_{BE}$) and the polarization parameter of the 90°-mirror in the telescope ($a_0$). Explanation for the parameters are given in the text. The configuration is given for SALTRACE-3 (summer 2014).

| Property | 355 nm | | 532 nm | | 1064 nm | |
|---|---|---|---|---|---|---|
| | Value | Uncertainty | Value | Uncertainty | Value | Uncertainty |
| $\alpha$ | 0.9° | ± 0.2° | 1.1° | ± 0.1° | 0.9° | ± 0.2° |
| $D_E$ | 0.049 | ± 0.034 | 0.046 | ± 0.028 | 0.047 | ± 0.037 |
| $\Delta_E$ | 0° | ± 180° | 0° | ± 180° | 0° | ± 180° |
| $\beta$ | 0° | ± 1° | 0° | ± 1° | 0° | ± 1° |
| $a_{BE}$ | 0.97 | ± 0.02 | 1.00 | -0.01 | 1.00 | - 0.01 |
| $D_0$ | 0.07 | ± 0.02 | -0.057 | ± 0.005 | 0.040 | ± 0.01 |
| $\Delta_0$ | 0° | ± 180° | 0° | ± 180° | 0° | ± 180° |
| $\gamma$ | 0° | ± 0.2° | 0° | ± 0.2° | 0° | ± 0.2° |
| $a_0$ | 1.00 | -0.01 | 0.958 | ± 0.005 | 1.00 | -0.01 |
| $D_T$ | 0.07 | ± 0.01 | 0.088 | ± 0.01 | 0.080 | ± 0.01 |
| $D_R$ | -0.903 | ± 0.01 | -0.992 | ± 0.01 | -0.980 | ± 0.01 |
| $\nu_R$ | 0° | ± 0.5° | 0° | ± 0.5° | 0° | ± 0.5° |

**Table 4.** The results of the simulation of the lidar system for the volume linear depolarization ratio. The simulation has been performed for the theoretical Rayleigh background and a typical value for the dust layer for each wavelength. The maximum and minimum value for each simulation and its standard deviation is given.

| Wavelength | Rayleigh | | | | Dust layer | | | |
|---|---|---|---|---|---|---|---|---|
| | input | min | max | std | input | min | max | std |
| 355 nm | 0.0080 | -0.0130 | 0.0440 | 0.0995 | 0.080 | 0.056 | 0.118 | 0.010 |
| 532 nm | 0.0053 | -0.0020 | 0.0260 | 0.0049 | 0.200 | 0.188 | 0.224 | 0.006 |
| 1064 nm | 0.0036 | -0.0050 | 0.0250 | 0.0057 | 0.230 | 0.214 | 0.257 | 0.007 |

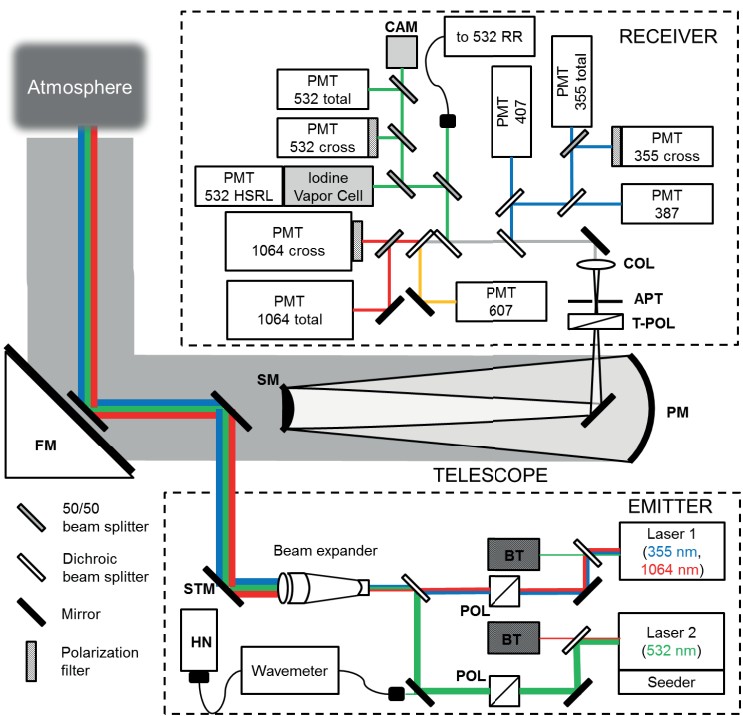

**Figure 1.** Sketch of BERTHA's emitter and receiver. All detection channels are photomultiplier tubes (PMT, operated in the photon counting mode), the number indicate the central wavelength of transmission in nanometer of the interference filter. Polarization filters perpendicular oriented to the emitted state of polarization are placed in front of the 'cross' channels, the 'total' channels measure the cross and the parallel part of the backscattered light, POL denotes polarizer to purify the laser polarization, BT beam trap, HN helium neon laser as reference for the wavemeter, STM steering mirror with stepper motors to adjust the overlap, CAM camera to visualize the overlap, FM flat mirror, consisting of two mirrors, the small one for emitted pulses, the large one for backscattered light, respectively, PM stands for main or primary mirror and SM for counter or secondary mirror of the Cassegrain telescope, TPOL means turnable polarization filter for $\Delta$ 90° calibration, APT motorized aperture, and COL collimator. Finally RR denotes rotational Raman channel, and HSRL High Spectral Resolution Lidar channel.

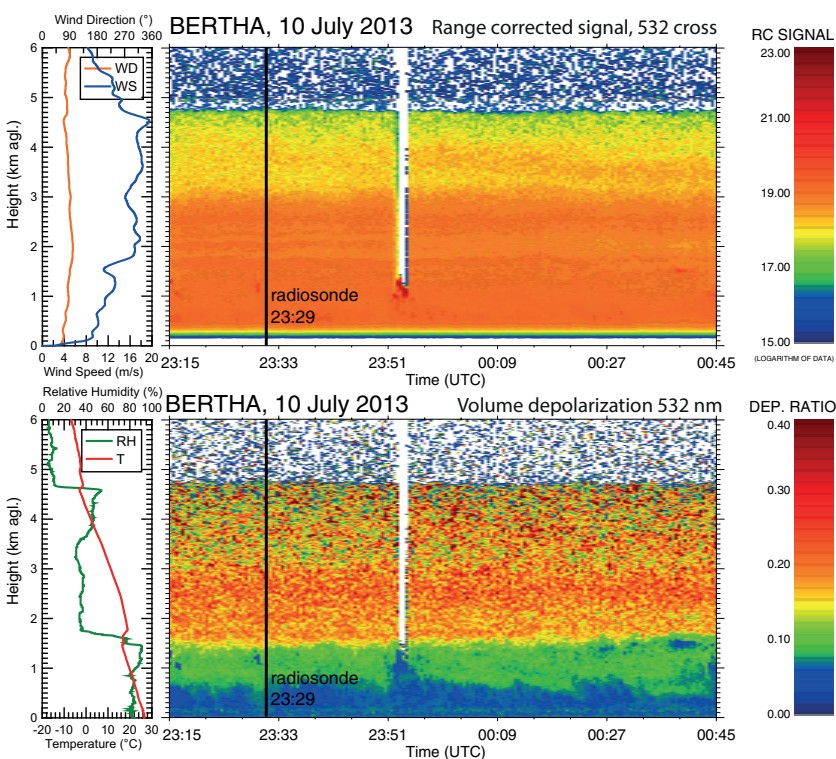

**Figure 2.** Saharan air layer (from 1.5–4.7 km height above ground level) above the marine boundary layer. The upper right panel shows the range-corrected cross-polarized 532 nm backscatter signal with temporal and vertical resolution of 30 s and 7.5 m, respectively. The lower right panel shows the 532 nm volume linear depolarization ratio. The lidar observation was performed on 10 July 2013, 19:15–20:45 local time. A radiosonde was launched at 19:29 local time (indicated by black vertical lines). The radiosonde profiles of wind speed (WS) and wind direction (WD) are shown in the upper left panel, the profiles of relative humidity (RH) and temperature (T) in the lower left panel.

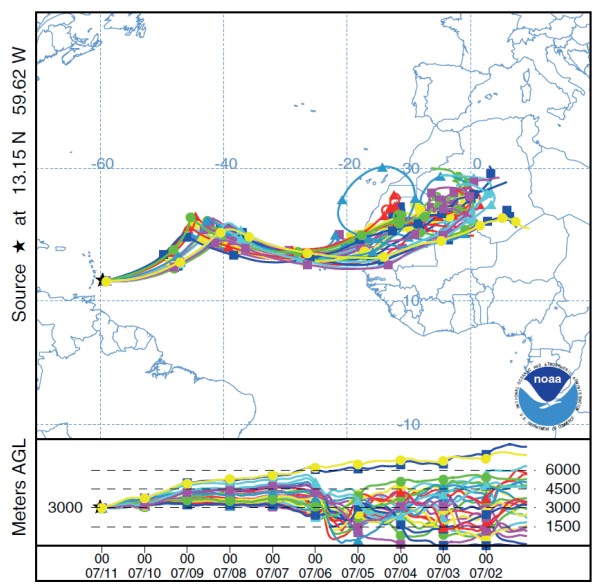

**Figure 3.** Ensemble of 10-day backward trajectories (Stein et al., 2015; HYSPLIT, 2016) for 11 July 2013, 0100 UTC, arriving at at 3000 m over Barbados.

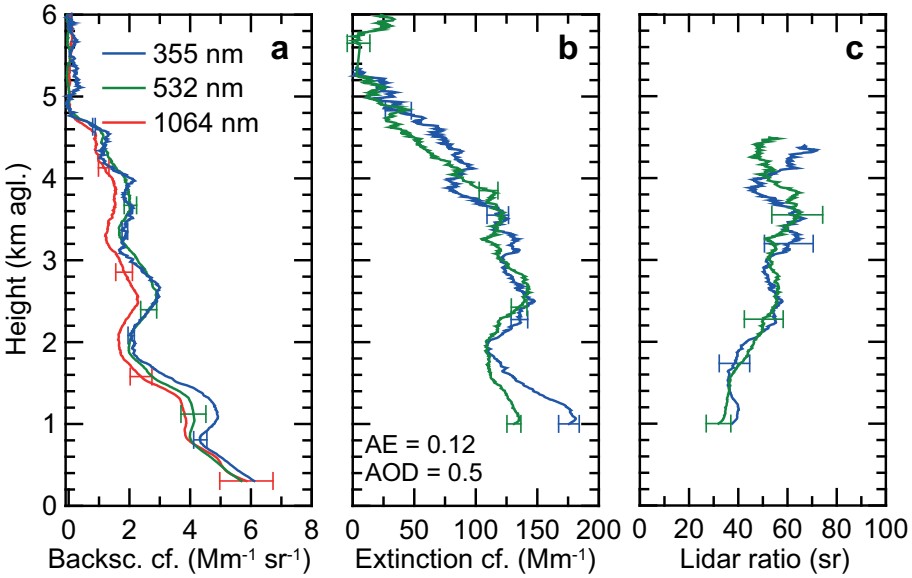

**Figure 4.** 45-minute mean profiles of (a) the particle backscatter coefficient at three wavelengths, (b) extinction coefficient at two wavelengths, and (c) extinction-to-backscatter ratio (lidar ratio) at two wavelengths, measured with BERTHA on 11 July 2013, 00:00–00:45 UTC. Error bars indicate the retrieval uncertainty (one standard deviation). The vertical signal smoothing window length is 200 m (backscatter coefficient) and 1000 m (extinction coefficient, lidar ratio). The column values of the Ångström exponent (AE 440-870 nm) and the aerosol optical depth (AOD) from the closest AERONET observation are indicated.

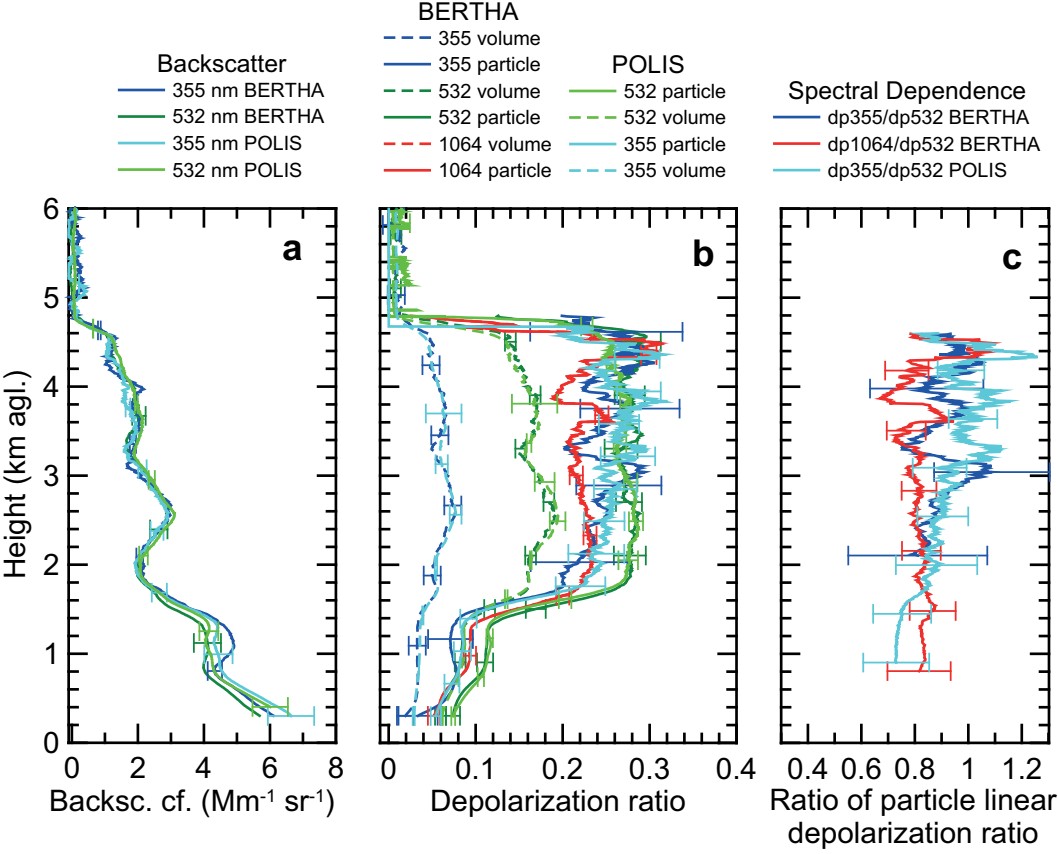

**Figure 5.** Comparison of POLIS and BERTHA depolarization-ratio observations. Shown are 45-minute mean profiles of (a) the particle backscatter coefficient at 355 (POLIS light blue, BERTHA dark blue) and 532 nm wavelength (POLIS light green, BERTHA dark green), (b) volume (dashed curves) and particle linear depolarization ratio (solid lines) at 355 nm (POLIS light blue, BERTHA dark blue), 532 nm (POLIS light green, BERTHA dark green), and 1064 nm (BERTHA, red line, equal volume and particle depolarization ratio profiles), (c) ratio of the 355 nm to 532 nm particle depolarization ratio (POLIS light blue, BERTHA dark blue) and 1064 nm to 532 nm particle depolarization ratio (BERTHA, red). POLIS and BERTHA observations were taken simultaneously on 11 July 2013, 00:00–00:45 UTC. Error bars indicate the retrieval uncertainty (one standard deviation). The vertical signal smoothing window length is 200 m (BERTHA, POLIS).

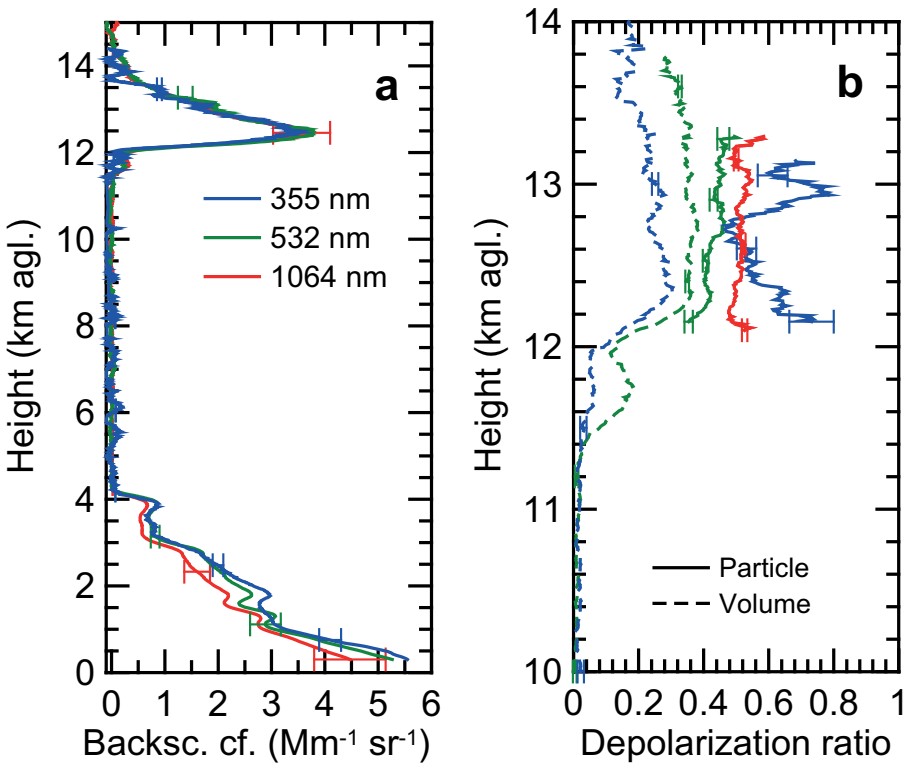

**Figure 6.** (a) Aerosol layers below 4 km height and cirrus layer from 12–14 km height in terms of particle backscatter coefficient at three wavelengths observed with lidar on 20–21 June 2014, 2310-0210 UTC (signal averaging period), and (b) cirrus ice crystal depolarization ratio (solid lines, volume depolarization ratio as dashed lines) at three wavelengths. At 1064 nm the volume depolarization ratio is equal to the particle depolarization ratio. The vertical signal smoothing length is 200 m. Error bars show the relative uncertainty in the retrievals (one standard deviation).

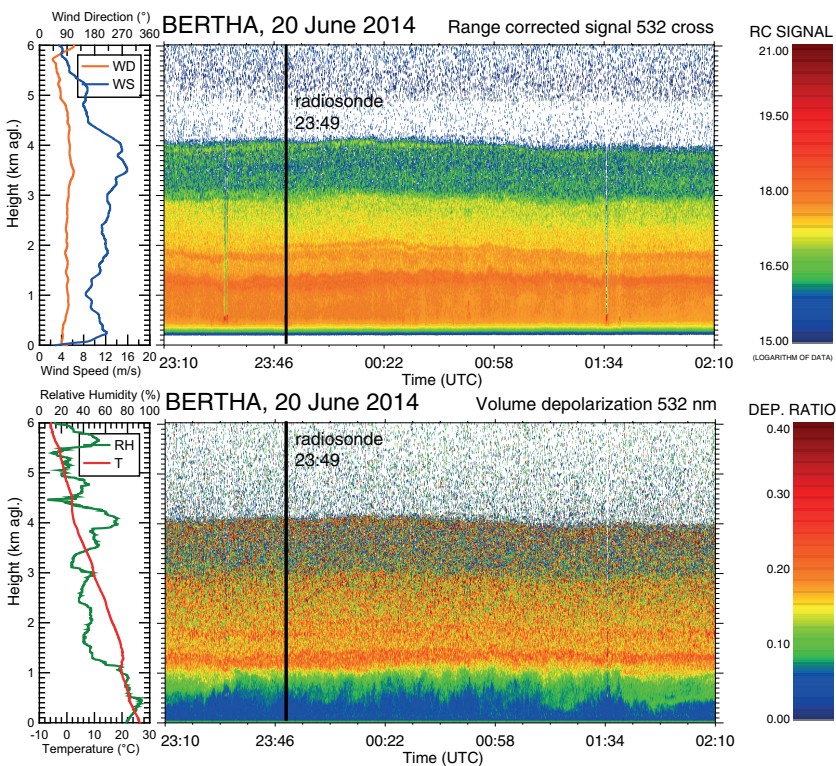

**Figure 7.** Saharan air layer (from 1–4 km height) above the marine boundary layer. The same parameters as in Fig. 2 are shown. The lidar observation was performed on 20 June 2014, 19:10–22:10 local time. The radiosonde was launched at 19:46 local time (indicated by black vertical lines).

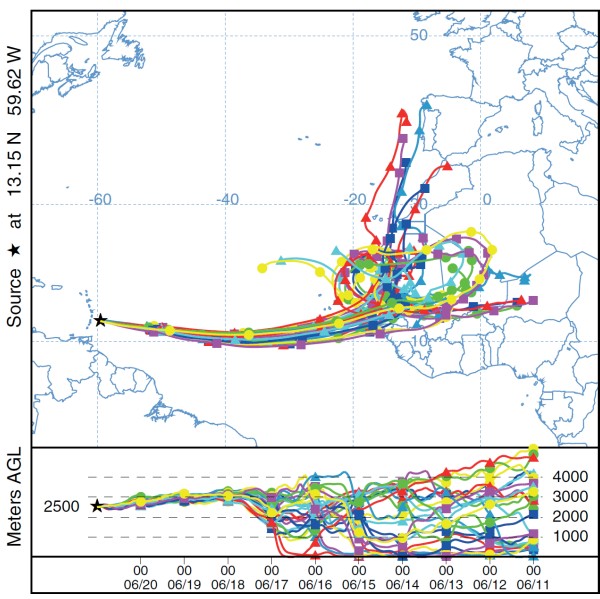

**Figure 8.** Ensemble of 10-day HYSPLIT backward trajectories for 21 June 2014, 00:00 UTC, arriving at at 2500 m over Barbados.

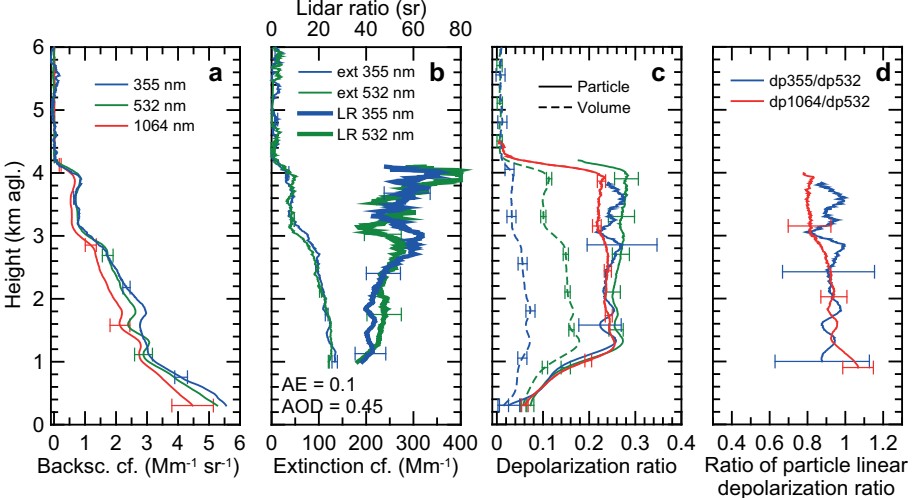

**Figure 9.** 3-hour mean profiles of (a) the particle backscatter coefficient at three wavelengths, (b) extinction coefficient and lidar ratio at two wavelengths, (c) volume and particle linear depolarization ratio at three wavelengths, and (d) ratio of the 355 nm to 532 nm and 1064 nm to 532 nm particle depolarization ratio. The lidar observation was performed on 20–21 June 2014, 23:10–02:10 UTC. The vertical signal smoothing window length is 200 m (backscatter coefficient, depolarization ratio) and 750 m (extinction coefficient, lidar ratio). Error bars indicate the retrieval uncertainty.

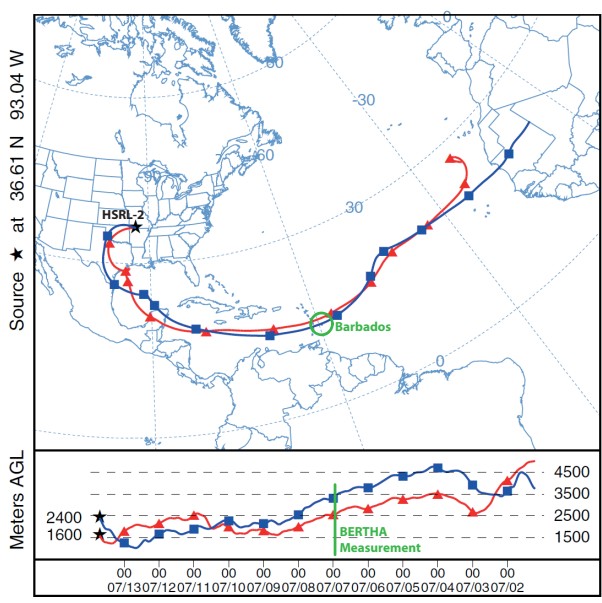

**Figure 10.** 12.5-day HYSPLIT backward trajectories for 13 July 2014, 17:00 UTC, arriving at at 1600 m (red) and 2400 m (blue) over northern Missouri (Midwestern United States). The location (Barbados) and time of the corresponding BERTHA lidar measurement is indicated by a green vertical line.

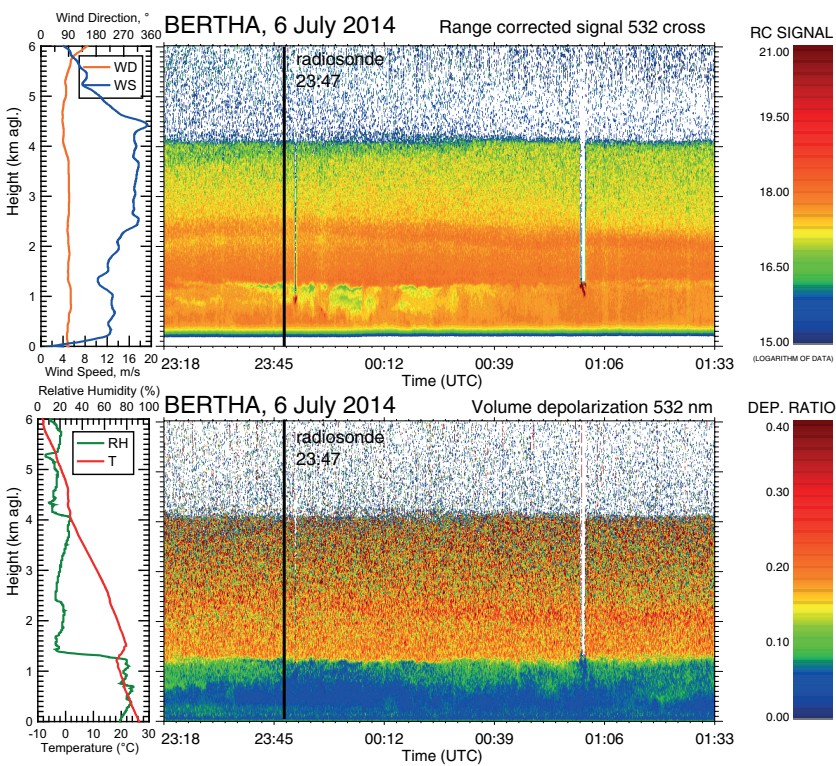

**Figure 11.** Saharan air layer (from 1.2–4.2 km height) above the marine boundary layer. The same parameters as in Fig. 2 are shown. The lidar observation was performed on 6 July 2014, 19:18–21:33 local time. The radiosonde was launched at 19:47 local time (indicated by a black vertical line).

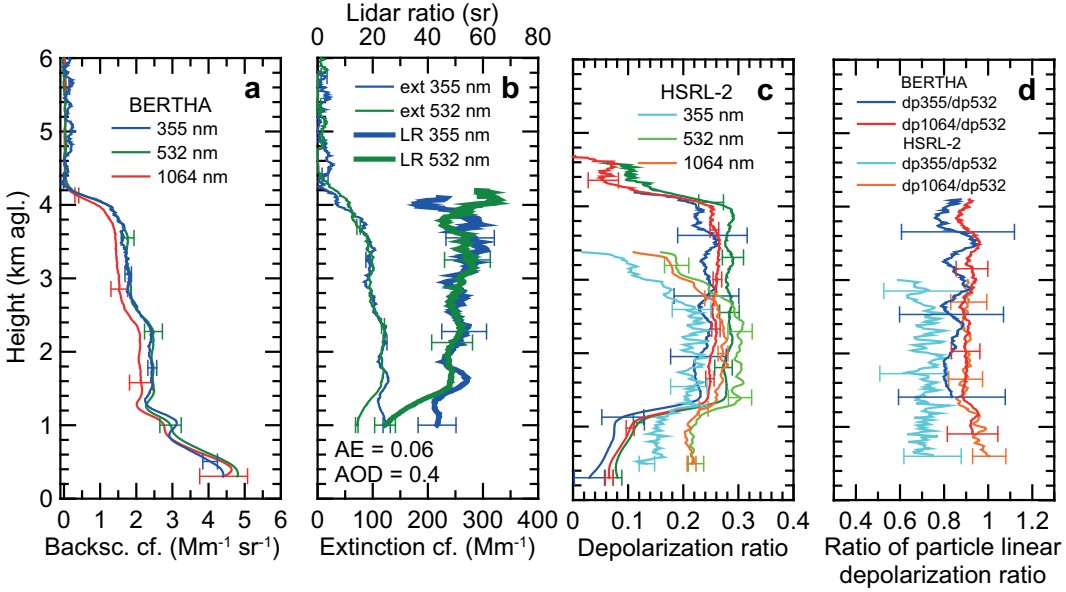

**Figure 12.** Same as Fig. 9, except for 6–7 July 2014, 23:18–01:33 UTC. For comparison, respective height profiles of the particle linear depolarization ratio (355 nm in light blue, 532 nm in light green, and 1064 nm in orange and for the ratio of depolarization ratios (in light blue and orange) measured with an airborne triple-wavelength polarization lidar (HSRL-2) (Burton et al., 2015) on 13 July 2014, 1700 UTC are shown. The airborne lidar observations were performed in Missouri (Midwestern United States), about 7000 km and 7 days downwind of Barbados.

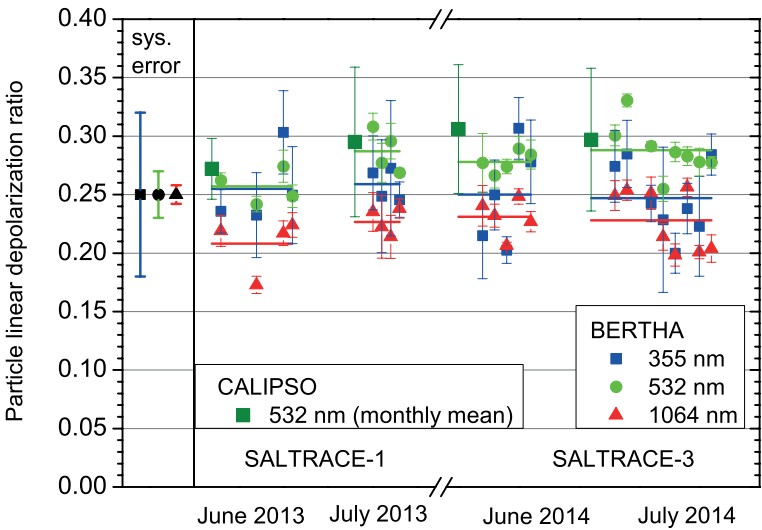

**Figure 13.** Overview of SAL layer mean particle depolarization ratios measured with triple-wavelength polarization lidar BERTHA during SALTRACE-1 (2013) and SALTRACE-3 (2014). Only triple-wavelength observations (one per evening) are considered. Horizontal lines show the mean values of all measurements conducted in one of the four SALTRACE months in 2013 and 2014. Error bars indicated the standard deviation calculated from all depolarization values (available with 50 m vertical resolution) between SAL base and top. Respective CALIOP monthly mean values and standard deviations of the SAL column particle depolarization ratios at 532 nm are shown for comparison. Level 2 version 3.30 CALIPSO Aerosol profile data around Barbados are used for this retrieval (CALIPSO, 2016). The systematic error bars shown on the left-hand side illustrate the impact of all the made retrieval assumptions on the results.

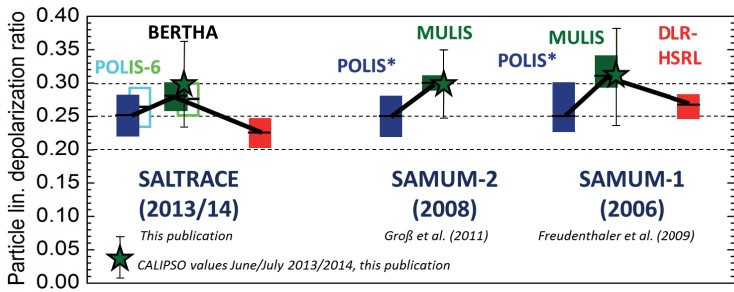

**Figure 14.** Comparison of dust-layer mean particle linear depolarization ratios measured during SAMUM-1 (Morocco), SAMUM-2 (Cabo Verde), and SALTRACE (Barbados). Colored bars show the range of mostly observed depolarization ratios at 355 nm (blue), 532 nm (green) and 1064 nm (red). The lidars BERTHA, POLIS, MULIS (the second polarization lidar of Munich University), and the airborne High Spectral Resolution Lidar (HSRL) of DLR (Deutsches Zentrum für Luft- und Raumfahrt) were used to collect this data set. Data are taken from the publications of (Freudenthaler et al., 2009) (SAMUM-1, 19 cases for MULIS, less for the other systems (2-4)), (Groß et al., 2011a) (SAMUM-2, 9 cases for POLIS, 5 cases for MULIS), (Groß et al., 2015) (SALTRACE, POLIS-6) and from this study (21 cases). In addition, CALIOP mean values of SAL column depolarization ratios considering all observations during the four SALTRACE months (June and July in 2013 and 2014) are shown. The mean values consider all CALIOP overpasses of selected areas in southeastern Morocco, in the Cabo Verde region, and around Barbados during these four months.

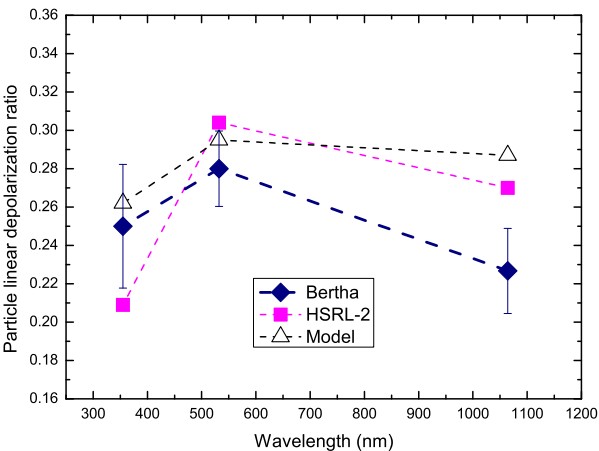

**Figure 15.** SALTRACE mean dust particle linear depolarization ratios at 355, 532, and 1064 nm (SAL column mean values and corresponding standard deviations) measured with BERTHA (21 cases). In addition, the HSRL-2 observation (Burton et al., 2015)) discussed in Sect. 3.3 is shown. The observational findings are compared with respective model results which are based on the dust size-shape characteristics described in Gasteiger et al. (2017).

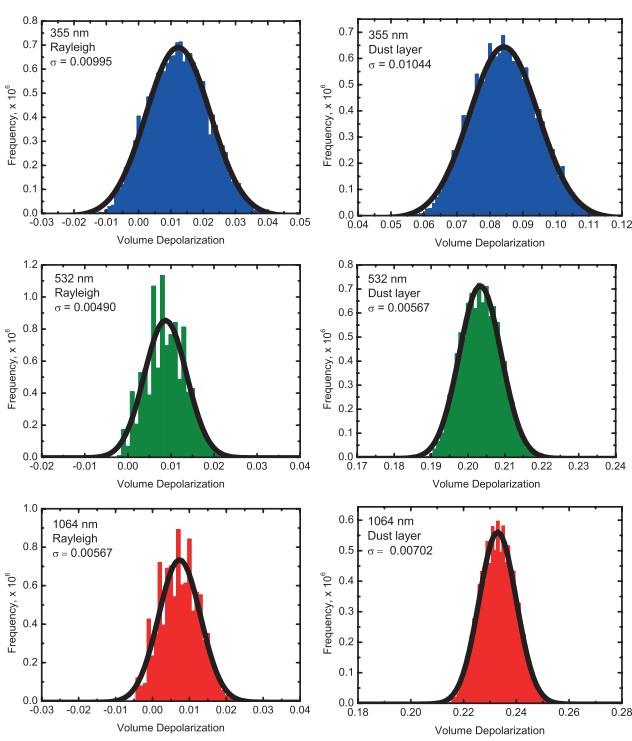

**Figure 16.** Frequency distribution (in millions of individual computations) of the volume depolarization ratio derived by a Monte-Carlo simulation of the system parameters of the BERTHA lidar system (Tab. 3) within their uncertainties. For each wavelength the Rayleigh depolarization and a typical value in the dust layer was chosen as the input value (true value), given in Table 4. The standard deviation $\sigma$ of the Gaussian fit (black line) is given to estimate the error of the volume depolarization ratio.