# Peer review of "Triple-wavelength depolarization-ratio profiling of Saharan dust over Barbados during SALTRACE in 2013 and 2014"

_Atmospheric Chemistry and Physics, 2017_

## Referee Comment (RC1) · Anonymous Referee #1 · 31 Mar 2017

The paper of Haarig et al. presents a study on the spectral dependency of the lidar-derived depolarization for dust aerosols. This is a very interesting aspect that has already been covered in other publications in the same group, but which deserves to be published under the SALTRACE program. It also provides additional information on the evolution of dust depolarization during long range transport over the tropical Atlantic. Whereas the tools used are well presented and referenced, there are some important aspects of the article that need to be clarified before publication. It is also necessary to review some parts of the results presentation, especially in section 3.

Major remarks:

Abstract

[Figure]

SALTRACE-2 does not have to appear in the abstract because it is not used.

Introduction

The introduction is well constructed and it clearly shows the interest of lidar systems with several polarization channels. The democratization of lidar systems of this type has an interest in the broader atmospheric physics domain, but this can only be done if manpower and money are available, which is far to be the case everywhere. Moreover, there are limitations related to eye safety, to obtain authorizations from air navigation authorities, regardless of location in the world. It would be nice to write a few words about it. About the dust study, please expand the reference list to other groups than the team of the authors. Many others have carried out and published scientific works on the subject of dust.

Section 2

The SALTRACE project is very interesting and has been conducted with a relevant approach.

Line 34p6: Why "is not optimized"? Hopefully, there have been many uncertainty studies on BERTHA and it must have a better SNR than POLIS.

This section refers to Appendix A.

Appendix A

This appendix is quite long, but the results are coming from another article. The calibration process presented is fairly standard in optics. It is laborious and indeed necessary when broad interferential filters (some nm) are used, which integrate temperature sensitive of highly depolarized inelastic scattering lines. But it has been published several times before.

Conversely, it is necessary to give clear information on the stability over time of the calibration. It can evolve with different environmental factors (for example the cleanliness on the optics, the temperature, etc.) and with the aging of certain elements of the detection chain.

Remark: the Mueller matrix computation usually requires precise information on materials and optics in general. That does not seem so easy to obtain. Nevertheless, even if the values are not perfectly accurate, this does not hamper the cited sensitivity study.

Section 3

This section should be reorganized in order to make it easier for the reader to understand its aim. Between 3 and 3.1, explain that there are 3 case studies and how these different cases are interesting.

Sub-section 3.1 Rename the title to 1st case study: ...

Lines 14-15p8: What parameters are homogeneous? If you are talking about the AOT, this is not the case (see MODIS).

Lines 18-19p8: This is high for a MBL and actually there can be local effects. The MBL is rather blue in Fig. 2 and the layer above is in the free troposphere. The last one can be created by forcing at the MBL top, linked to the presence of Sc for example. It may be necessary to look at CALIPSO measurements to confirm or disprove this.

Lines 24-25p8 (fig. 3): With 10-day backtrajectories, it is more relevant to use the ensemble mode of Hysplit.

Line 27p8: Provide a reference or explanation for what is called the Âń classical Raman lidar technique Âż.

Line 1p9: There are previous references to marine aerosols, for example Flamant et al., JGR, 1998.

Lines 4-5p9: Why not use what has been found in Fig. 4 to inverse POLIS? BERTHA could also be inverted with Gross et al. (2015) and compared to the previous inversion without using POLIS in order to minimize the polarization calibration effect of both

lidars. POLIS seems redundant in this section.

Lines 20-24p9: The error bars overlap and it seems difficult to conclude with certainty.

In Fig.4, are the error bars including atmospheric variability during the averaging time?

Sub-section 3.2 (instead of 3.1.1) (remove)

This section is more of a discussion and should not be included here. Moreover, it is based only on previous work. It is more a report of the state of the art that has its place in an introduction. The set of values presented comes from Mamouri and Ansmann (2017). It is not clear how this applies to the data used in the article.

Lines 1-5p10: The aerosol mode ratio changes a lot if you look at AERONET and the values given here can significantly change.

Lines 20-25p10: This is a conclusion, but not for this article.

Sub-section 3.2 (stay section 3.2) The title is not in the topic of the paper. Change the title to 2nd case study: . . .

Lines 27p10-7p11: This is an important point to check the consistency of the depolarization data between the 3 wavelengths of the BERTHA lidar. This should be placed in a dedicated paragraph, before presenting the case studies. Why did not you evaluate the LRs on your measurements themselves?

Line 25p11: Could you explain why?

Sub-section 3.3 Change the title of sub-section 3.3 to: "3rd case study: dust transport from Africa to Arkansas over 12000 km"

Section 4

In all the text, the meteorological descriptions encountered during the campaigns must be given earlier in the article, for instance after sub-section 2.1.

Lines2-3p14: The sentence is not clear.

Line 21p14: The differences between these values are in the error bar, thus, the differences are not significant. For the comparison between SALTRACE, SAMUM1 and SAMUM2, are you certain that the dust sources are the same and that the aerosol nature does not change?

Lines 1-5p15: there is no change between 532 and 1064 nm from the simulations, why?

Minor remarks:

Line 17p3: lidarstudies to lidar studies Line 18p3: assuarance to assurance Line 30p6: remove V before et al. Line 21p8: 532 nm instead of 1064 nm? Line 33p9: 50-60% is it the fine mode fraction? Line 12p10: What observations? Line 17p10: diameter in the... Line 17p11: replace equal by very close Lines 32p10-2p12: Not useful for the paper. Lines 5 and 9p12: Cite the figures in order of appearance Line 7p13: remove "see sect. 2.4", not useful Lines 18-19p13: already said Line 20p17: amtmospheric to atmospheric Table 3: define each variable in the table caption.

---

## Referee Comment (RC2) · Anonymous Referee #2 · 11 Apr 2017

This paper deals with triple-wavelength depolarization ratio measurements performed in aged Saharan dust layers, highlighting the importance of multi wavelength polarization observations for dust modeling improvements and dust characterization. The authors rightly acknowledges previously reported studies, related to field/laboratory observations and to modeling simulations. The manuscript is well written, discussing interesting measurements, and worth being published in Atmospheric Chemistry and Physics, under the SALTRACE project. However, in order to be improved, I would suggest to the authors to take into consideration the following comments.

Please also note the supplement to this comment:

[Figure]

http://www.atmos-chem-phys-discuss.net/acp-2017-170/acp-2017-170-RC2-supplement.pdf

**Supplement:**

**General Comments:**

This paper deals with triple-wavelength depolarization ratio measurements performed in aged Saharan dust layers, highlighting the importance of multi wavelength polarization observations for dust modeling improvements and dust characterization. The authors rightly acknowledges previously reported studies, related to field/laboratory observations and to modeling simulations.

The manuscript is well written, discussing interesting measurements, and worth being published in Atmospheric Chemistry and Physics, under the SALTRACE project. However, in order to be improved, I would suggest to the authors to take into consideration the following comments.

**Major remarks:**

1. The introduction is well written providing a good overview. In my opinion in this section should be also mentioned that airborne triple-wavelength depolarization measurements of Saharan dust in Caribbean exist in the literature by Burton et al., 2015. However, this is something that is mentioned later in section 2 (Page 4, Line 33). Moreover, in this section, should be made clearer to the reader which is the focus of this manuscript.

2. Page 6, Line 34: Could you please comment briefly in which sense BERTHA was not optimized for depolarization observations?

3. Page 7, Line 15: To my understanding, for characterizing CALIOP observations as dust or not, a threshold value has been set regarding the particle depolarization ratio. Two lines later is mentioned that "only the observations characterized as dust or polluted dust from the CALIPSO…". In case that are used also scenes characterized as polluted dust, then lower values of particle depolarization ratios are introduced to the statistics. Could you please comment more on this and make it clearer?

4. The focus of the manuscript is to demonstrate triple-wavelength depolarization measurements of dust particles, through three case studies. These case studies have to be well organized and demonstrated in the manuscript, in order to be easier for the reader to follow up. Thus, I suggest to the authors to rename the sub-section headers accordingly. For example the 3.1 could be something like: "Case study I: 11 July 2013".

5. Page 8, Line 29: The authors are mentioning that the backscatter coefficient at 1064 nm is significantly lower than the corresponding values at 532 nm. This seems to be more intense during the first case study, compared to the next two. How could this be further explained? The temporal evolution of columnar Angstrom exponent (500/1020 nm) as obtained from the corresponding AERONET site, indicate a variability around 0.1-0.2. This seems to be in contradiction with the high backscatter related Angstrom exponent derived from 532 and 1064 nm (Fig. 4; around 0.5 in the SAL). Higher backscatter coefficient at 1064 nm, would lead to different spectral dependence of particle linear depolarization ratio at the wavelength pair of 1064 to 532 nm (Fig. 5c). By the way, I

think that it would be beneficial for the manuscript if for each case study, the corresponding columnar AERONET observations, are also presented.

6. The sub-section 3.1.1 is really interesting and provides valuable information for the findings presented in this study. However, in my opinion should be moved either in a separated paragraph in section 2 as a generic methodology followed during SALTRACE project to provide complementary dust related information, or has to be shorten in order to be directly included in the sub-section 3.1 which describes the Case I. Moreover, I think that the last paragraph (Page 10, Lines 20-25) of this sub-section is more related to the Conclusions section, than here.

7. As already mentioned (see major remark No. 4) you are kindly suggested to change the header of sub-section 3.2 to something like: "Case study II: 20-21 June 2014". Moreover, in my opinion a dedicated paragraph should be constructed earlier than section 3, in order to demonstrate the consistency of the depolarization data between the 3 wavelengths of the BERTHA lidar. This paragraph could include the comparison with POLIS and the cirrus case presented in Figures 5 and 6.

8. Why in Fig. 6b the volume depolarization profiles at 532 and 1064 nm are not plotted above the cirrus? Is this due to signal to noise issues of the corresponding cross-polarized channels? Moreover, if the volume depolarization ratio is equal to particle depolarization ratio at 1064 nm, this is something that should be mentioned in the caption of Fig. 6b.

9. In Fig. 14 it would be nice also to show the number of case studies used for the corresponding statistical values obtained with BERTHA, MULIS and POLIS during SALTRACE and SAMUM 1-2.

10. In my opinion Fig. 15 is an important figure summarizing all the scientific results and open questions of the manuscript. The scientific answers which are related to the less sharp (compared to observations) simulated dust depolarization spectral slopes, along with the discrepancies found in HSRL-2 and BERTHA observations (which are inverted when going to higher wavelengths) are scattered in the manuscript, but not well summarized when describing Fig. 15.

**Minor Remarks:**

1. Page 3, Line 17: Change "lidastudies" to "lidar studies".
2. Page 5, Lines 8-9: It is clear from Fig. 1 that the linear polarizers are behind the beam expanders however, you could mention this also in the manuscript. Moreover, it would be helpful if you could describe if there is a specific need for using two lasers in BERTHA setup? Is this due to the implemented HSRL channel at 532 nm, or is related to the plane of polarization of each laser generated harmonic?
3. Page 6, Line 30: "Freudenthaler, V. et al., 2016" delete the "V.".
4. Page 7, Line 31: I kindly suggest changing the AERONET site name with the one that can be found in AERONET website, namely Barbados_Saltrace.

5. Page 8, Line 21: In the manuscript, the wavelength of 1064 nm is mentioned while in Fig. 2 and its caption the 532 nm is shown. Please consider correcting where appropriately.

6. Page 8, Line 27: I would suggest replacing the text "classical Raman" with "conventional Raman".

7. Page 8, Line 28: Since there are many studies of Saharan dust observations with conventional Raman systems, I would suggest to the authors to provide some more references than "Tesche et al., 2011a".

8. Page 9, Line 18: Please consider providing same number of significant digits for common parameters. This is a general comment and I would suggest go over the entire manuscript and tables, in order to correct it where appropriately.

9. Page 9, Line 29: Delete the word "here" due to redundancy.

10. Page 11, Line 3: "BERTHA was operated at…this measurement". This information has been already provided earlier (page 5, line 10). I suggest you to delete it from here.

11. Page 13, Line 31: Change "give" to "given".

12. Page 17, Line 12: A definite article "the" is missing.

---

## Author Comment (AC1) · 24 Jul 2017

The comment was uploaded in the form of a supplement:
https://www.atmos-chem-phys-discuss.net/acp-2017-170/acp-2017-170-AC1-supplement.pdf

---

## Author Comment (AC2) · 24 Jul 2017

The Letter and the manuscript with the changes marked in bold can be found in the supplement. Thank you for the review.

Please also note the supplement to this comment: https://www.atmos-chem-phys-discuss.net/acp-2017-170/acp-2017-170-AC2-supplement.pdf

---

## Author Response (AR1)

**Letter of Reply to Referee 1**

Thank you for carefully reading the manuscript and providing useful suggestions to improve the paper. We want to summarize the main changes on the manuscript at the beginning and then reply to the specific remarks.

- **Section 4 contains only the statistical overview of the triple-wavelength depolarization observations of long-range transported dust.**
- **A new discussion section (Section 5) is introduced and contains the comparison to previous campaigns (formerly the later part of Section 4) and the discussion of the size information conveyed by the triple-wavelength depolarization measurements (formerly Section 3.1.1)**
- **A meteorological overview of the SALTRACE summer campaigns is added as new Section 2.2.**
- **A short historical overview of the polarization lidar observations of tropospheric dust with the relevant literature was added at the beginning of Section 2.3**
- **The polarization of the 90° mirror in the telescope has been found to have a significant influence on the 532 nm depolarization ratio, so the data and the error estimation has been recalculated. As a result, the 355 and 1064 nm depolarization ratio remained almost unchanged, but the 532 nm values are now decreased by about 0.02 (mean value from 0.30 to 0.28).**
- **The HSRL-2 measurement took place above Missouri and not Arkansas, as our coauthor S. Burton mentioned.**

**The changes in the manuscript are marked in bold except of Section 3 (case studies and former Section 3.1.1 now Page 14, Line 28 – Page 15 Line 25 in the discussion), where more things have changed.**

*Major remarks:*

*Abstract*

*SALTRACE-2 does not have to appear in the abstract because it is not used.*

**SALTRACE-2 is removed from the abstract.**

*Introduction*

*The introduction is well constructed and it clearly shows the interest of lidar systems with several polarization channels. The democratization of lidar systems of this type has an interest in the broader atmospheric physics domain, but this can only be done if manpower and money are available, which is far to be the case everywhere. Moreover, there are limitations related to eye safety, to obtain authorizations from air navigation authorities, regardless of location in the world. It would be nice to write a few words about it.*

**We never included such section in any of our papers during the last decades. So, we decided not to change our way to present our scientific products. So we leave out to discuss eye safety, air navigation, manpower and costs.**

*About the dust study, please expand the reference list to other groups than the team of the authors. Many others have carried out and published scientific works on the subject of dust.*

**We agree. The list is expanded (Page 2, Lines 6-15) by several other campaigns studying Saharan dust over Africa and the Atlantic Ocean.**

*Section 2*

*The SALTRACE project is very interesting and has been conducted with a relevant approach.*

*Line 34p6: Why "is not optimized"? Hopefully, there have been many uncertainty studies on BERTHA and it must have a better SNR than POLIS.*

**BERTHA is a complex lidar system, consisting of 13 receiver channels and several steering mirrors and evolved during the last 20 years step by step. So it is not designed from the beginning to be an excellent polarization lidar. Therefore an appendix for the polarization sensitive characterization of the system is included in this paper. But of course, BERTHA has a more powerful laser and a larger telescope leading to a better SNR. The sentence has been rephrased to make this clear on Page 7 Line 30.**

*This section refers to Appendix A.*

*Appendix A*

*This appendix is quite long, but the results are coming from another article. The calibration process presented is fairly standard in optics. It is laborious and indeed necessary when broad interferential filters (some nm) are used, which integrate temperature sensitive of highly depolarized inelastic scattering lines. But it has been published several times before.*

*Conversely, it is necessary to give clear information on the stability over time of the calibration. It can evolve with different environmental factors (for example the cleanliness on the optics, the temperature, etc.) and with the aging of certain elements of the detection chain.*

*Remark: the Mueller matrix computation usually requires precise information on materials and optics in general. That does not seem so easy to obtain. Nevertheless, even if the values are not perfectly accurate, this does not hamper the cited sensitivity study.*

**The appendix describes the specific properties of the triple-wavelength polarization Raman lidar system and the calculation of the error bars. A detailed description of the lidar system is the basis of the presented results and thus necessary to our opinion. The approach is not new, but published several times before, but it is now applied to the BERTHA lidar system. The broad interference filters are used among other reasons for geometrical reasons of the receiver.**

**The calibration procedure is described for the final state of the system at the end of the SALTRACE-3 campaign and afterwards back in Leipzig. The temperature of the container was kept constant by a strong air condition. But elements could have degraded over time. In order to see the stability over time, the Rayleigh depolarization in the background (usually between 6 -8 km height) was calculated and showed only slight variations over time. The variation was < 20% for 532 nm and < 30% for 355 nm.**

*Section 3*

*This section should be reorganized in order to make it easier for the reader to understand its aim. Between 3 and 3.1, explain that there are 3 case studies and how these different cases are interesting.*

**Thank you, a short explaining text at beginning of section 3 was added.**

*Sub-section 3.1 Rename the title to 1st case study: ...*

**Thank you for this comment. We changed the titles of the subsections to number and name the case studies.**

*Lines 14-15p8: What parameters are homogeneous? If you are talking about the AOT, this is not the case (see MODIS).*

**The sentence aimed to state the dust was distributed homogeneously with in the SAL and that the same structure was observed for several days. To not confuse the readers, the passage was deleted.**

*Lines 18-19p8: This is high for a MBL and actually there can be local effects. The MBL is rather blue in Fig. 2 and the layer above is in the free troposphere. The last one can be created by forcing at the MBL top, linked to the presence of Sc for example. It may be necessary to look at CALIPSO measurements to confirm or disprove this.*

**This is correct; the MBL does not extend up to 1.75 km. We distinguish between the (convective) marine boundary layer (MBL in blue) and the marine aerosol layer (MAL) which extends up to the trade wind inversion at around 1.75 km. In the absence of Saharan dust above, the MAL consists of marine aerosols only. This is the case in our winter measurements over Barbados (not shown in this paper). In summer the MAL is affected by downward mixing of dust particles. In other publications (for example Jung et al., JGR, 2013) three layers are distinguished: a subcloud layer (up to 500 m), a intermediate layer and the SAL. The passage has been rephrased to avoid misunderstandings (Page 9, Lines 19-21)**

*Lines 24-25p8 (fig. 3): With 10-day back trajectories, it is more relevant to use the ensemble mode of Hysplit.*

**We agree and changed Fig 3 and 8 accordingly.**

*Line 27p8: Provide a reference or explanation for what is called the "classical Raman lidar technique"*

**It is changed to "conventional Raman technique" and a reference for calculating the extinction and backscatter coefficient independently by using the vibrational Raman lines was added (Page 9, Line 30).**

*Line 1p9: There are previous references to marine aerosols, for example Flamant et al., JGR, 1998.*

**Five more references for the marine lidar ratio were added (Flamant et al., 1998; Burton et al., 2012; Dawson et al., 2015; Rittmeister et al., 2017; Haarig et al., 2017) at Page 10, Lines 5-6.**

*Lines 4-5p9: Why not use what has been found in Fig. 4 to inverse POLIS? BERTHA could also be inverted with Gross et al. (2015) and compared to the previous inversion without using POLIS in order to minimize the polarization calibration effect of both lidars. POLIS seems redundant in this section.*

**Using the Klett or Raman method to retrieve backscatter and extinction does not affect the volume depolarization ratio, which is the most important to compare the quality of the depolarization measurements. For the particle depolarization ratio, the backscatter coefficient has indeed an influence. The comparison with Polis is shown to demonstrate the accuracy of the depolarization ratios measured with BERTHA, as Polis is an EARLINET reference system with high accuracy (see Freudenthaler's contribution to the 27th ILRC, Freudenthaler, EJP, 2016).**

*Lines 20-24p9: The error bars overlap and it seems difficult to conclude with certainty.*

**Sure, but we have POLIS and all the comparison point to the same spectral dependence as we found. Figure 5 is the case study that shows it rather clearly.**

*In Fig.4, are the error bars including atmospheric variability during the averaging time?*

**The atmospheric variability is not included in the error bars.**

*Sub-section 3.2 (instead of 3.1.1) (remove)*

*This section is more of a discussion and should not be included here. Moreover, it is based only on previous work. It is more a report of the state of the art that has its place in an introduction. The set of values presented comes from Mamouri and Ansmann (2017). It is not clear how this applies to the data used in the article.*

**We agree that this section fits better to the discussion and moved it to Section 5 (discussion), where the spectral behavior of the depolarization ratio is discussed (Page 14, Line 28 – Page 15 Line 25).**

*Lines 1-5p10: The aerosol mode ratio changes a lot if you look at AERONET and the values given here can significantly change.*

**The FMF can vary from day to day (from dust outbreak to dust outbreak). But we do not see these variations within one plume event. We checked always the AERONET observations (before sunset or**

**briefly after sunrise) closest to our lidar observations, and we did not see any large variability in the FMF.**

*Lines 20-25p10: This is a conclusion, but not for this article.*

**It is moved with the section to the discussion and modified slightly.**

*Sub-section 3.2 (stay section 3.2) The title is not in the topic of the paper. Change the title to 2nd case study: . . .*

**The title is changed and the case studies are numbered.**

*Lines 27p10-7p11: This is an important point to check the consistency of the depolarization data between the 3 wavelengths of the BERTHA lidar. This should be placed in a dedicated paragraph, before presenting the case studies. Why did not you evaluate the LRs on your measurements themselves?*

**Thank you for your suggestion, but we decided to demonstrate the consistency of our three-wavelength depolarization measurements in the presence of a cirrus in this case study. It was the second night measurement of the third SALTRACE campaign, and so it is important to show, that no significant changes in the system occurred within one year. The cirrus was too thin to derive good Raman measurements of the extinction with a meaningful smoothing length. In Section 2.3 a short paragraph was added concerning the cirrus to check the consistency of the depolarization ratios (Page 7, Lines 14-18).**

*Line 25p11: Could you explain why?*

**Most probably the dust has a different transport history, now described at Page 11, Lines 10-13. The depolarization ratios for the upper layer (3-4 km) and the lower layer (1.4-3 km) are within the observed variability for dust arriving on Barbados (see Fig. 13),**

*Sub-section 3.3 Change the title of sub-section 3.3 to: "3rd case study: dust transport from Africa to Arkansas over 12000 km"*

**The title is changed and the case studies are numbered.**

*Section 4*

*In all the text, the meteorological descriptions encountered during the campaigns must be given earlier in the article, for instance after sub-section 2.1.*

**Thank you for your suggestion. A dedicated subsection 2.2 after the description of the SALTRACE projected was added to describe the meteorological situation.**

*Lines2-3p14: The sentence is not clear.*

**The sentence was reshaped (Page 13, Lines 27-31). The 1064-nm depolarization ratio is more sensitive to the large particles, whose fraction might change during long-range transport.**

*Line 21p14: The differences between these values are in the error bar, thus, the differences are not significant. For the comparison between SALTRACE, SAMUM1 and SAMUM2, are you certain that the dust sources are the same and that the aerosol nature does not change?*

**The difference in 532nm depolarization ratio is not significant. A note concerning this was added on Page 14, Line 15. We assume that the aerosol source are always the same (west Sahara source) and do not change with time. All our results agree with this assumptions reasonable well.**

*Lines 1-5p15: there is no change between 532 and 1064 nm from the simulations, why?*

**The preliminary model calculations based on reasonable shape parametrizations and a rough consideration of the size distribution. The particle coarse mode is probably overestimated in the model. The model is used up to size parameter 10 only, for size parameter > 10 an estimation is necessary. The paragraph Page 15, Lines 30-33, has been reshaped to express the uncertainties in the model assumptions.**

**Minor remarks:**

*Line 17p3: lidarstudies to lidar studies*
**Thank you. It is changed.**

*Line 18p3: assuarance to assurance*

**Thank you. It is changed.**

*Line 30p6:remove V before et al.*

**Removed.**

*Line 21p8: 532 nm instead of 1064 nm?*

**Thank you, indeed the 532 nm volume depolarization ratio is shown.**

*Line 33p9: 50-60%is it the fine mode fraction?*

**Yes, it is the fine mode fraction. It is stated more precisely now.**

*Line 12p10: What observations?*

**Now, that this paragraph is placed in the discussion, it gets clearer. The observations are the spectral depolarization ratio measurements of long-range transported dust.**

*Line 17p10: diameter inthe. . .*

**Thank you, it is corrected.**

*Line 17p11: replace equal by very close*

**Replaced.**

*Lines 32p10-2p12: Not useful for the paper.*

**We deleted the paragraph and recommend studying Mamouri and Ansmann, 2017**

*Lines 5 and 9p12: Cite the figures in order of appearance*

**Thank you, we changed the order of figures.**

*Line 7p13: remove"see sect. 2.4", not useful*

**Removed.**

*Lines 18-19p13: already said*

**But to our opinion it is good to say it at this part again.**

*Line 20p17: amtmospheric to atmospheric*

**Changed, thank you.**

*Table 3: define each variable in the table caption.*

**The variables are now defined in the caption of the table to make it easier for the reader.**

**Letter of Reply to Referee 2**

**Thank you for carefully reading the manuscript and providing useful suggestions to improve the paper. We want to summarize the main changes on the manuscript at the beginning and then reply to the specific remarks.**

- **Section 4 contains only the statistical overview of the triple-wavelength depolarization observations of long-range transported dust.**
- **A new discussion section (Section 5) is introduced and contains the comparison to previous campaigns (formerly the later part of Section 4) and the discussion of the size information conveyed by the triple-wavelength depolarization measurements (formerly Section 3.1.1)**
- **A meteorological overview of the SALTRACE summer campaigns is added as new Section 2.2.**
- **A short historical overview of the polarization lidar observations of tropospheric dust with the relevant literature was added at the beginning of Section 2.3**
- **The polarization of the 90° mirror in the telescope has been found to have a significant influence on the 532 nm depolarization ratio, so the data and the error estimation has been recalculated. As a result, the 355 and 1064 nm depolarization ratio remained almost unchanged, but the 532 nm values are now decreased by about 0.02 (mean value from 0.30 to 0.28).**
- **The HSRL-2 measurement took place above Missouri and not Arkansas, as our coauthor S. Burton mentioned.**

**The changes in the manuscript are marked in bold except of Section 3 (case studies and former Section 3.1.1 now Page 14, Line 28 – Page 15 Line 25 in the discussion), where more things have changed.**

General Comments:

*This paper deals with triple-wavelength depolarization ratio measurements performed in aged Saharan dust layers, highlighting the importance of multi wavelength polarization observations for dust modeling improvements and dust characterization. The authors rightly acknowledge previously reported studies, related to field/laboratory observations and to modeling simulations.*

*The manuscript is well written, discussing interesting measurements, and worth being published in Atmospheric Chemistry and Physics, under the SALTRACE project. However, in order to be improved, I would suggest to the authors to take into consideration the following comments.*

*Major remarks:*

*1. The introduction is well written providing a good overview. In my opinion in this section should be also mentioned that airborne triple-wavelength depolarization measurements of Saharan dust in Caribbean exist in the literature by Burton et al., 2015. However, this is something that is mentioned later in section 2 (Page 4, Line 33). Moreover, in this section, should be made clearer to the reader which is the focus of this manuscript.*

**You are right; the focus of the paper is strengthened by adding a paragraph in the introduction (Page 3, Lines 21-24). It is good to mention the Burton study already in the introduction. We add it on Page 3, Lines 19-21.**

*2. Page 6, Line 34: Could you please comment briefly in which sense BERTHA was not optimized for depolarization observations?*

**BERTHA is a complex lidar system, consisting of 13 receiver channels and several steering mirrors and evolved during the last 20 years step by step. So it is not designed from the beginning to be an excellent polarization lidar. Therefore an appendix for the polarization sensitive characterization of the system is included in this paper. The sentence has been rephrased to make this clear on Page 7 Line 30.**

*3. Page 7, Line 15: To my understanding, for characterizing CALIOP observations as dust or not, a threshold value has been set regarding the particle depolarization ratio. Two lines later is mentioned that "only the observations characterized as dust or polluted dust from the CALIPSO…". In case that are used also scenes characterized as polluted dust, then lower values of particle depolarization ratios are introduced to the statistics. Could you please comment more on this and make it clearer?*

**We thank the reviewer for this comment. We re-calculated CALIPSO means using only the observations characterized as dust. The new CALIPSO means are higher by 0.01 above Barbados and Cape Verde, while above Morocco are identical. We change the corresponding text in the manuscript: Page 8, Line 11: "only the observations characterized as dust from the CALIPSO"**

*4. The focus of the manuscript is to demonstrate triple-wavelength depolarization measurements of dust particles, through three case studies. These case studies have to be well organized and demonstrated in the manuscript, in order to be easier for the reader to follow up. Thus, I suggest to the authors to rename the sub-section headers accordingly. For example the 3.1 could be something like: "Case study I: 11 July 2013".*

**Thank you for this comment. We changed the headers of the subsections to number and name the case studies.**

*5. Page 8, Line 29: The authors are mentioning that the backscatter coefficient at 1064 nm is significantly lower than the corresponding values at 532 nm. This seems to be more intense during the first case study, compared to the next two. How could this be further explained? The temporal evolution of columnar Angstrom exponent (500/1020 nm) as obtained from the corresponding AERONET site, indicate a variability around 0.1-0.2. This seems to be in contradiction with the high backscatter related Angstrom exponent derived from 532 and 1064 nm (Fig. 4; around 0.5 in the SAL). Higher backscatter coefficient at 1064 nm, would lead to different spectral dependence of particle linear depolarization ratio at the wavelength pair of 1064 to 532 nm (Fig. 5c). By the way, I think that it would be beneficial for the manuscript if for each case study, the corresponding columnar AERONET observations, are also presented.*

**Thank you for your helpful comment. We carefully checked all our data, especially the reference values for the 1064 nm backscatter. As a result we updated Fig. 4. Additionally, the corresponding AERONET observations are now given for all case studies in Fig. 4, 9 and 12.**

*6. The sub-section 3.1.1 is really interesting and provides valuable information for the findings presented in this study. However, in my opinion should be moved either in a separated paragraph in section 2 as a generic methodology followed during SALTRACE project to provide complementary dust related information, or has to be shorten in order to be directly included in the sub-section 3.1 which describes the Case I. Moreover, I think that the last paragraph (Page 10, Lines 20-25) of this sub-section is more related to the Conclusions section, than here.*

**We moved this subsection to the discussion (section 5, Page 14, Line 28 – Page 15 Line 25), because it discusses our three-wavelength depolarization observations and sets them in a broader context.**

*7. As already mentioned (see major remark No. 4) you are kindly suggested to change the header of sub-section 3.2 to something like: "Case study II: 20-21 June 2014". Moreover, in my opinion a dedicated paragraph should be constructed earlier than section 3, in order to demonstrate the consistency of the depolarization data between the 3 wavelengths of the BERTHA lidar. This paragraph could include the comparison with POLIS and the cirrus case presented in Figures 5 and 6.*

**The subsection header was changed. We decided to not construct a separate paragraph to show the consistency of our depolarization data, but to use the case studies to demonstrate this. To check the results with a reference system or to use a cirrus case, is a normal tool that should be used always and thus is part of a case study. In this way the case studies give an idea about the vertical structure of dust layer over Barbados and indicate the quality of our depolarization measurements. In Section 2.3 a short paragraph was added concerning the cirrus to check the consistency of the depolarization ratios (Page 7, Lines 14-18).**

*8. Why in Fig. 6b the volume depolarization profiles at 532 and 1064 nm are not plotted above the cirrus? Is this due to signal to noise issues of the corresponding cross-polarized channels? Moreover, if the volume depolarization ratio is equal to particle depolarization ratio at 1064 nm, this is something that should be mentioned in the caption of Fig. 6b.*

**The volume depolarization ratio becomes noisy above the cirrus and was therefore not plotted. A sentence to the volume and particle depolarization ratio at 1064 nm was added to the caption of Fig 6b.**

*9. In Fig. 14 it would be nice also to show the number of case studies used for the corresponding statistical values obtained with BERTHA, MULIS and POLIS during SALTRACE and SAMUM 1-2.*

**We went through the reference and added the information. For SAMUM 1 and 2, there were not so many cases. The information is added in the caption of Fig. 14.**

*10. In my opinion Fig. 15 is an important figure summarizing all the scientific results and open questions of the manuscript. The scientific answers which are related to the less sharp (compared to observations)*

*simulated dust depolarization spectral slopes, along with the discrepancies found in HSRL-2 and BERTHA observations (which are inverted when going to higher wavelengths) are scattered in the manuscript, but not well summarized when describing Fig. 15.*

**Thank you for pointing this out. By moving Section 3.1.1 to the discussion and discuss the spectral slope in this separate Section, the focus is strengthened in Section 5.**

*Minor Remarks:*

*1. Page 3, Line 17: Change "lidastudies" to "lidar studies".*

**Thank you. It is changed.**

*2. Page 5, Lines 8-9: It is clear from Fig. 1 that the linear polarizers are behind the beam expanders however, you could mention this also in the manuscript. Moreover, it would be helpful if you could describe if there is a specific need for using two lasers in BERTHA setup? Is this due to the implemented HSRL channel at 532 nm, or is related to the plane of polarization of each laser generated harmonic?*

**The comment on the position of the linear polarizer was added (Page 5, Line 29). In fact a linear polarizer after the beam expander would be a good idea, but it is technically challenging because of a beam diameter of around 10 cm. Two lasers are used for two reasons, firstly to have a frequency stabilized 532-nm laser for the implementation of the HSRL-channel and secondly to have a backup laser in the field campaign (Page 5, Lines 32-33).**

*3. Page 6, Line 30: "Freudenthaler, V. et al., 2016" delete the "V.".*

**Deleted.**

*4. Page 7, Line 31: I kindly suggest changing the AERONET site name with the one that can be found in AERONET website, namely Barbados_Saltrace.*

**The site name is indicated (Page 8, Line 26)**

*5. Page 8, Line 21: In the manuscript, the wavelength of 1064 nm is mentioned while in Fig. 2 and its caption the 532 nm is shown. Please consider correcting where appropriately.*

**The text has been changed to "532 nm".**

*6. Page 8, Line 27: I would suggest replacing the text "classical Raman" with "conventional Raman".*

**It is changed to "conventional Raman" and a reference is provided (Page 9, Line 30).**

*7. Page 8, Line 28: Since there are many studies of Saharan dust observations with conventional Raman systems, I would suggest to the authors to provide some more references than "Tesche et al., 2011a".*

**We agree; there are many observations. The reference list has been extended by Mattis et al., 2002; Papayannis et al., 2005; Tesche et al., 2011a; Preißler et al., 2011; Veselovskii et al., 2016; Hofer et al., 2017 (Page 9, Line 31 – Page 10, Line 1). But there are many more Raman lidar observations of dust.**

*8. Page 9, Line 18: Please consider providing same number of significant digits for common parameters. This is a general comment and I would suggest go over the entire manuscript and tables, in order to correct it where appropriately.*

**Thank you, we went through the manuscript and unified the number of significant digits. For the particle linear depolarization ratio we use normally 2 significant digits (for an overall average 3), for the volume depolarization ratio in the Rayleigh background we need 4 digits.**

*9. Page 9, Line 29: Delete the word "here" due to redundancy.*

**Deleted.**

*10. Page 11, Line 3: "BERTHA was operated at…this measurement". This information has been already provided earlier (page 5, line 10). I suggest you to delete it from here.*

**The passage has been deleted.**

*11. Page 13, Line 31: Change "give" to "given".*

**Changed.**

*12. Page 17, Line 12: A definite article "the" is missing.*

**It was added, thank you.**

[revised manuscript text omitted]